# SE(3) Diffusion Model-based Point Cloud Registration for Robust 6D Object Pose Estimation

**Haobo Jiang[1], Mathieu Salzmann[2,3]\*, Zheng Dang[2], Jin Xie[1]\*, and Jian Yang[1]\***

[1]PCA Lab, Nanjing University of Science and Technology, China
[2]CVLab, EPFL, Switzerland
[3]ClearSpace, Switzerland
`{jiang.hao.bo, csjxie, csjyang}@njust.edu.cn`
`{zheng.dang, mathieu.salzmann}@epfl.ch`

## Abstract

In this paper, we introduce an SE(3) diffusion model-based point cloud registration framework for 6D object pose estimation in real-world scenarios. Our approach formulates the 3D registration task as a denoising diffusion process, which progressively refines the pose of the source point cloud to obtain a precise alignment with the model point cloud. Training our framework involves two operations: An SE(3) diffusion process and an SE(3) reverse process. The SE(3) diffusion process gradually perturbs the optimal rigid transformation of a pair of point clouds by continuously injecting noise (perturbation transformation). By contrast, the SE(3) reverse process focuses on learning a denoising network that refines the noisy transformation step-by-step, bringing it closer to the optimal transformation for accurate pose estimation. Unlike standard diffusion models used in linear Euclidean spaces, our diffusion model operates on the SE(3) manifold. This requires exploiting the linear Lie algebra $\mathfrak{se}(3)$ associated with SE(3) to constrain the transformation transitions during the diffusion and reverse processes. Additionally, to effectively train our denoising network, we derive a registration-specific variational lower bound as the optimization objective for model learning. Furthermore, we show that our denoising network can be constructed with a surrogate registration model, making our approach applicable to different deep registration networks. Extensive experiments demonstrate that our diffusion registration framework presents outstanding pose estimation performance on the real-world TUD-L, LINEMOD, and Occluded-LINEMOD datasets. The code is available at https://github.com/Jiang-HB/DiffusionReg.

## 1 Introduction

Accurately estimating the 6D pose of an object, comprising both its spatial position and orientation, is a critical task in the field of computer vision, and has been extensively applied in various domains such as robotics grasping [10, 52, 68], augmented reality [32, 33], and autonomous navigation [7, 16, 58]. While substantial efforts have been dedicated to developing methods for 6D pose estimation based on RGB or RGB-D data [26, 42, 41, 52, 38, 63, 49], the advancements of 3D sensors (such as Kinect and LiDAR) and 3D registration techniques has promoted the emergence of point cloud registration-based pose estimation as a promising direction.

---

\*Corresponding authors.

Haobo Jiang, Jin Xie, and Jian Yang are with PCA Lab, Key Lab of Intelligent Perception and Systems for High-Dimensional Information of Ministry of Education, and Jiangsu Key Lab of Image and Video Understanding for Social Security, School of Computer Science and Engineering, Nanjing University of Science and Technology, China.

Nevertheless, the state-of-the-art methods for object-level 3D registration [44, 62, 15] primarily focus on synthetic data and struggle to yield precise registration on real-world 6D pose estimation datasets such as TUD-L [20], LINEMOD [18], and Occluded-LINEMOD [6]. Unlike the controlled geometric structures, manually defined transformations, and artificial partial overlap present in synthetic data, real-world pose estimation datasets present significant challenges, including large rotations and translations, natural noise interference, and severe occlusions, considerably exacerbating the registration difficulty. Prior attempts to tackling these challenges include [11], which introduces a general match normalization layer to regularize the feature distribution of the source and model point clouds thus facilitating feature matching, and [60, 48], which design RANSAC-based outlier rejection strategy to enhance the registration robustness. While these initial attempts constitute significant steps towards making object-level point cloud registration practical, much work remains to be done to fully address all the aforementioned real-world challenges.

In this paper, inspired by the remarkable success of diffusion models in generative AI [19, 45, 46, 47], we propose to adapt them to the 3D registration field, and introduce an SE(3) diffusion model-based registration method for robust 6D object pose estimation in the real world. Our approach formulates 3D registration as a denoising diffusion process in SE(3) (special Euclidean group), aiming to progressively refine the pose of the source point cloud to achieve a precise alignment with the model point cloud. Training our model then involves two key operations: An SE(3) diffusion process and an SE(3) reverse process. The SE(3) diffusion process progressively converts the optimal transformation between the source and model point clouds into a noise one by continuously injecting perturbation transformations. Conversely, the SE(3) reverse process learns a denoising network to gradually refine the noise transformation to the optimal one. Our diffusion model operates on the SE(3) manifold, which poses the challenge of extending the diffusion/reverse formulas from linear Euclidean space to a nonlinear manifold. To address this challenge, we exploit the linear Lie algebra $\mathfrak{se}(3)$ associated with SE(3) to conduct linear diffusion/reverse computations and map the $\mathfrak{se}(3)$ results to SE(3) to obtain the desired diffused/reverse transformations.

To effectively train our denoising network, we exploit a Bayesian formalism and derive a registration-specific variational lower bound as optimization objective for model optimization. Furthermore, we reformulate our denoising network with a surrogate registration model, which makes our approach applicable to different deep registration methods such as [54, 62]. During the inference stage, the learned denoising network progressively refines an identity transformation to approach the optimal one, given the source and model point clouds as conditioning signals.

Compared to previous registration methods, our diffusion registration framework offers the following two advantages. First, the diffusion process generates a diverse set of poses for the source point cloud, allowing for more comprehensive model training. This increased pose diversity facilitates the model's ability to handle a wider range of rotation/translation and improves its generalization capabilities. Second, by incorporating the Bayesian posterior in the reverse process, our approach effectively guides the update of the source point cloud pose at each reverse step. This guidance mitigates the risk of getting trapped in local optima, leading to more robust and accurate pose estimation results. Our empirical results provide substantial evidence supporting these two advantages.

To summarize, our main contributions are as follows: **(i)** We introduce a novel SE(3) diffusion model-based 3D registration framework for robust 6D object pose estimation, where the optimal transformation is estimated via a progressive denoising process. **(ii)** To train our denoising network, we follow a Bayesian approach and establish a 3D registration-specific variational lower bound as optimization objective for our SE(3) diffusion model. Furthermore, we reformulate the denoising network with a surrogate registration model, enabling the integration of different deep registration models into our framework. **(iii)** To the best of our knowledge, we are the first to successfully adapt the diffusion model from linear Euclidean space to the SE(3) point cloud registration task for 6D object pose estimation. Our extensive experiments on real-world datasets, including TUD-L [20], LINEMOD [18] and Occluded-LINEMOD [6], confirm the effectiveness of our framework.

## 2    Related Work

**Point Cloud Registration.** Point cloud registration aims to estimate the rigid transformation between a pair of point clouds. Existing methods can be roughly divided into optimization-based techniques and deep learning-based ones. Iterative Closest Point (ICP) [4] alternately searches for the closest correspondences and estimates transformation until convergence. Go-ICP [61] uses the branch-and-bound algorithm to improve the robustness of ICP to initialization. Robust ICP [64] designs a

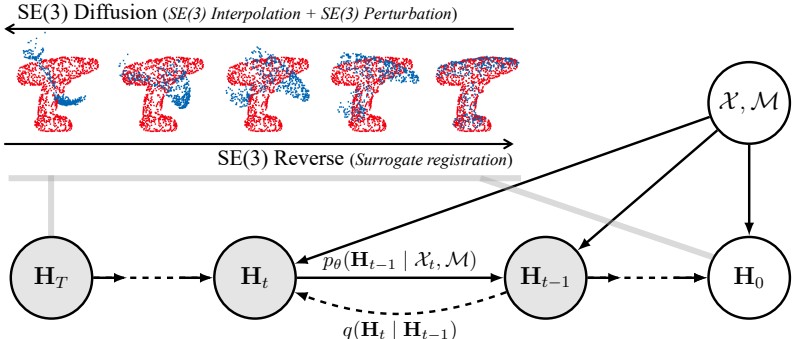

SE(3) Diffusion (*SE(3) Interpolation + SE(3) Perturbation*)

SE(3) Reverse (*Surrogate registration*)

$p_\theta(\mathbf{H}_{t-1} \mid \mathcal{X}_t, \mathcal{M})$

$q(\mathbf{H}_t \mid \mathbf{H}_{t-1})$

Figure 1: Probabilistic graphical model of our SE(3) diffusion model-based registration framework.

Welsch's function-based robust metric to obtain an outlier-robust alignment evaluation for optimization. Sparse ICP [5] reformulates ICP using sparsity-inducing norms. Many other ICP variants have been proposed, such as[8, 43, 14, 2, 17], exhibiting promising registration performance. Nevertheless, the current main research effort focuses on deep learning models. In this context, DCP [54] constructs pseudo correspondences for SVD-based transformation estimation using deep feature similarity. PRNet [55] leverages keypoint identification and Gumbel softmax for establishing more reliable correspondences. PointNetLK [1] and FMR [21] introduce the Lucas & Kanada (LK) algorithm [3] and inverse compositional (IC) algorithm [3] into deep models for transformation refinement via feature alignment. CEMNet [23] uses a planning-based cross-entropy method for transformation optimization. RPMNet [62], RGM [15] and RIENet [44] leverage Sinkhorn optimization, neighborhood structure consistency, and graph matching, respectively, for robust outlier removal. Many other deep registration models such as [28, 9, 36, 30, 29, 67] have been developed and achieve impressive registration performance. However, these registration models mainly focus on synthetic datasets, such as ModelNet40 [56]. As shown in [11], they still yield limited precision on real-world 6D object pose estimation datasets such as TUD-L [20], LINEMOD [18], and Occluded-LINEMOD [6].

**6D Object Pose Estimation.** Estimating the 6D pose (orientation and position) of objects has gained increasing attention in recent years. Early approaches directly estimated object pose from extracted image features through regression or classification. PoseCNN [57] decouples 6D pose estimation into center-based translation regression and quaternion-based rotation regression. SSD-6D [24] extends the single-shot object detector to cover the full 6D object space for pose inference. Trabelsi *et al.* [51] propose a multi-attentional network for iterative pose refinement using the appearance and flow information. DeepIM [31] and LatentFusion [37] learn pose estimation by minimizing the error between the rendered model and the observation. More recent research has focused on a two-stage estimation pipeline, where 2D keypoints are extracted to solve for 6D pose using the PnP algorithm [13]. Rad *et al.* [42] take the predicted 2D projections of the corners of 3D bounding boxes as the keypoints, while Zhao *et al.* [65] manually designate keypoints over the surface of the object model. PVNet[41] establishes a pixel-wise voting network for keypoint estimation using predicted pixel-wise voting vectors. Other methods [6, 40, 50, 52, 49, 52] also present promising pose estimation performance. Recently, thanks to the advances in 3D registration techniques, the community has witnessed a growing interest in 3D registration-based 6D object pose estimation, which recovers the object's 6D pose by estimating the rigid transformation between the object (source) and the model point clouds. This is what we achieve here, but, in contrast to most existing methods which focus on synthetic data, we tackle the challenging scenario of working with real point clouds.

## 3 Approach

### 3.1 Revisiting the Euclidean Diffusion Model

Diffusion models, as generative models, aim to generate new data by progressively denoising noisy inputs [19, 45, 46, 47]. Their training phase involves a diffusion process and a reverse process. The diffusion process successively injects Gaussian noise into the data sample $\mathbf{x}_0 \sim p_{data}$ ($p_{data}$ indicates the distribution of the training data) so as to gradually transform $\mathbf{x}_0$ into the noise data $\mathbf{x}_T \sim \mathcal{N}(\mathbf{0}, \mathbf{I})$ (standard Gaussian distribution), thus forming a Markov chain $\mathbf{x}_0 \rightarrow \mathbf{x}_1 \rightarrow \cdots \rightarrow \mathbf{x}_T$. As demonstrated in [19], the random variable $\mathbf{x}_t \sim q(\mathbf{x}_t \mid \mathbf{x}_{t-1}) := \mathcal{N}(\mathbf{x}_t; \sqrt{1-\beta_t}\mathbf{x}_{t-1}, \beta_t\mathbf{I})$ can also be expressed in a closed form $\mathbf{x}_t \sim q(\mathbf{x}_t \mid \mathbf{x}_0)$, which can be formulated as follows:

$$\mathbf{x}_t = \sqrt{\bar{\alpha}_t}\mathbf{x}_0 + \sqrt{1-\bar{\alpha}_t}\boldsymbol{\varepsilon}, \ \boldsymbol{\varepsilon} \sim \mathcal{N}(\mathbf{0}, \mathbf{I}) \ , \tag{1}$$

Here, the diffusion coefficients $\bar{\alpha}_t = \prod_{s=0}^t \alpha_s = \prod_{s=0}^t (1 - \beta_s)$, and $\beta_s$ indicates the noise coefficient determined by a linear schedule [19] or a cosine schedule [35]. Then, the reverse process learns a denoising network, a parameterized normal distribution $p_\theta(\mathbf{x}_{t-1} \mid \mathbf{x}_t) := \mathcal{N}(\mathbf{x}_{t-1}; \boldsymbol{\mu}_\theta(\mathbf{x}_t, t), \beta_t \mathbf{I})$, to progressively denoise the noisy data $\mathbf{x}_T$ into the clean one $\mathbf{x}_0$, forming a reverse Markov chain $\mathbf{x}_T \rightarrow \mathbf{x}_{T-1} \rightarrow \cdots \rightarrow \mathbf{x}_0$. Here, $\boldsymbol{\mu}_\theta(\mathbf{x}_t, t)$ indicates the parameterized mean of the normal distribution. The following variational lower bound of the log likelihood over the training data is then derived as the optimization objective for training the denoising network:

$$\mathbb{E}_{\mathbf{x}_0 \sim p_{data}}[\log p_\theta(\mathbf{x}_0)] \geq \mathbb{E}_q[\sum_{t>1} D_{KL}(q(\mathbf{x}_{t-1} \mid \mathbf{x}_t, \mathbf{x}_0) \| p_\theta(\mathbf{x}_{t-1} \mid \mathbf{x}_t)) - \log p_\theta(\mathbf{x}_0 \mid \mathbf{x}_1)]. \quad (2)$$

Based on Bayes' formula, the random variable $\mathbf{x}_{t-1}$ following the posterior distribution $q(\mathbf{x}_{t-1} \mid \mathbf{x}_t, \mathbf{x}_0)$ in Eq. 2 can be represented as

$$\mathbf{x}_{t-1} = \frac{\sqrt{\bar{\alpha}_{t-1}} \beta_t}{1 - \bar{\alpha}_t} \mathbf{x}_0 + \frac{\sqrt{\alpha_t}(1 - \bar{\alpha}_{t-1})}{1 - \bar{\alpha}_t} \mathbf{x}_t + \sqrt{\tilde{\beta}_t} \boldsymbol{\varepsilon}, \quad (3)$$

where the random variable $\boldsymbol{\varepsilon} \sim \mathcal{N}(\mathbf{0}, \mathbf{I})$ and the variance scale $\tilde{\beta}_t = \frac{1 - \bar{\alpha}_{t-1}}{1 - \bar{\alpha}_t} \beta_t$.

### 3.2 SE(3) Diffusion Registration Model for 6D Object Pose Estimation

In the context of 6D object pose estimation based on point clouds, our objective is to determine the rigid transformation between a partially-scanned source point cloud $\mathcal{X} = \{\mathbf{x}_i \in \mathbb{R}^3\}_{i=1}^N$ and a complete model point cloud $\mathcal{M} = \{\mathbf{m}_j \in \mathbb{R}^3\}_{j=1}^M$ so as to align their overlapping regions precisely. This transformation encompasses a rotation matrix $\mathbf{R} \in SO(3)$ and a translation vector $\mathbf{t} \in \mathbb{R}^3$, representing the orientation of the object and its spatial position, respectively. To obtain a partial source point cloud, following [11], we mask an input depth map to restrict it to the object, and convert the resulting depth values into 3D points using the known camera intrinsic parameters. By contrast, the model point cloud is generated by uniformly sampling a complete mesh model. We denote the optimal transformation $\mathbf{H}_0 \in SE(3)$ and the identity transformation $\mathbb{H} \in SE(3)$ as

$$\mathbf{H}_0 = \begin{bmatrix} \mathbf{R} & \mathbf{t} \\ \mathbf{0}^\top & 1 \end{bmatrix}, \quad \mathbb{H} = \begin{bmatrix} \mathbf{I} & \mathbf{0} \\ \mathbf{0}^\top & 1 \end{bmatrix}, \quad (4)$$

both of which are represented by $4 \times 4$ homogeneous transformation matrices.

Inspired by the progressive generation of diffusion models, we adopt the concept of denoising diffusion to tackle the point cloud registration task, specifically in the SE(3) space, and thus propose an SE(3) diffusion model-based 3D registration framework for robust 6D object pose estimation. While [27, 53] have introduced variations of the diffusion model on SE(3) or SO(3) manifolds, they are not suitable for our 3D registration task. The training phase of our framework thus involves an SE(3) diffusion process and an SE(3) reverse process. Given a pair of source and model point clouds, the former continuously disturbs their optimal transformation by injecting noise into it, while the latter aims to learn a denoising network to progressively convert the noisy transformation to the optimal one. As such, during inference, the learned denoising network can be leveraged to recover the transformation between the source and model point clouds through a progressive denoising process. The training phase and inference phase of our algorithm are detailed in Algorithms 1 and 2. Below, we provide a detailed explanation of our SE(3) diffusion process and SE(3) reverse process.

#### 3.2.1 SE(3) Diffusion Process

In accordance with the standard diffusion model described in Sec. 3.1, our SE(3) diffusion process progressively disturbs the optimal transformation $\mathbf{H}_0$ of point clouds $\mathcal{X}$ and $\mathcal{M}$ into a noisy transformation $\mathbf{H}_T$ by injecting perturbation transformations (noise) into it, thus forming a diffusion Markov chain: $\mathbf{H}_0 \rightarrow \mathbf{H}_1 \rightarrow \cdots \rightarrow \mathbf{H}_T$. However, our SE(3) diffusion process features two critical differences from the conventional one: **(i)** Our diffusion process operates on the nonlinear SE(3) manifold, unlike the standard diffusion process which acts in linear Euclidean space. Consequently, the linear diffusion operations provided in Eq 1 cannot be directly applied to our model. **(ii)** The standard diffusion process assumes the noise variable $\mathbf{x}_T \sim \mathcal{N}(\mathbf{0}, \mathbf{I})$ to be centered around the zero vector. However, in our model for 3D registration, we require the noise transformation $\mathbf{H}_T$ to be centered around the identity transformation $\mathbb{H}$. To account for these differences while generating the

| **Algorithm 1** Training phase | **Algorithm 2** Inference phase |
|---|---|
| **repeat** | **for** $t = T, ..., 1$ **do** |
|    Sample $\mathcal{X}, \mathcal{M}, \mathbf{H}_0 \sim p_{data}$ |    Register $\hat{\mathbf{H}}_{t \to 0} = f_\theta(\mathcal{X}_t, \mathcal{M})$, where $\mathcal{X}_t =$ |
|    Sample $t \sim \text{Uniform}(\{1, ..., T\})$, $\mathbf{H}_t =$ |    $\mathcal{T}(\mathcal{X}, \mathbf{H}_t)$ |
|    $\text{Exp}(\gamma\sqrt{1 - \bar{\alpha}_t}\varepsilon)\mathcal{F}(\sqrt{\bar{\alpha}_t}; \mathbf{H}_0, \mathbb{H})$ |    Estimate $\mathbf{H}_{t-1} = \text{Exp}(\lambda_0 \text{Log}(\hat{\mathbf{H}}_{t \to 0}\mathbf{H}_t) +$ |
|    Register $\hat{\mathbf{H}}_{t \to 0} = f_\theta(\mathcal{T}(\mathcal{X}, \mathbf{H}_t), \mathcal{M})$ |    $\lambda_1 \text{Log}(\mathbf{H}_t))$ |
|    Optimize loss $\mathcal{L}_t = \text{loss}(\hat{\mathbf{H}}_{t \to 0}, \mathbf{H}_0\mathbf{H}_t^{-1})$ | **end for** |
| **until** converged | **Return**: $\mathbf{H}_0$ |

desired diffusion Markov chain, we propose to interpolate the transformations between $\mathbf{H}_0$ and $\mathbb{H}$ on the SE(3) manifold, incorporating noise perturbations. Formally, our interpolation-based SE(3) diffusion formula formulates $\mathbf{H}_t \sim q(\mathbf{H}_t \mid \mathbf{H}_0)$ at time step $t$ $(1 \le t \le T)$ as

$$\mathbf{H}_t = \underbrace{\text{Exp}(\gamma\sqrt{1 - \bar{\alpha}_t}\varepsilon)}_{\text{Perturbation}} \underbrace{\mathcal{F}(\sqrt{\bar{\alpha}_t}; \mathbf{H}_0, \mathbb{H})}_{\text{Interpolation}}, \; \varepsilon \sim \mathcal{N}(\mathbf{0}, \mathbf{I}), \tag{5}$$

In essence, this formalism utilizes an interpolation function $\mathcal{F}(\sqrt{\bar{\alpha}_t}; \mathbf{H}_0, \mathbb{H})$ to derive an intermediate transformation that lies between the optimal transformation $\mathbf{H}_0$ and the identity transformation $\mathbb{H}$. Subsequently, we augment this interpolated transformation with a randomly-sampled perturbation transformation (noise), denoted as $\text{Exp}(\gamma\sqrt{1 - \bar{\alpha}_t}\varepsilon)$, to yield the diffused transformation $\mathbf{H}_t$. Below, we will introduce these two components in detail.

**Transformation Interpolation Function** $\mathcal{F}(\sqrt{\bar{\alpha}_t}; \mathbf{H}_0, \mathbb{H})$. The interpolation function $\mathcal{F} : SE(3) \times SE(3) \times [0, 1] \to SE(3)$ aims to compute an intermediate transformation between the optimal transformation $\mathbf{H}_0$ and the identity one $\mathbb{H}$. Following the setting of diffusion coefficients in Eq. 1, we set the interpolation weight on transformation $\mathbf{H}_0$ at time step $t$ to $\sqrt{\bar{\alpha}_t}$ and, consequently, the interpolation weight on $\mathbb{H}$ is $1 - \sqrt{\bar{\alpha}_t}$. As the time step $t$ increases, the interpolation weight $\sqrt{\bar{\alpha}_t}$ on $\mathbf{H}_0$ decreases gradually, resulting in a transition of the interpolated transformation from $\mathbf{H}_0$ to $\mathbb{H}$. However, due to the non-linearity of the SE(3) manifold, direct linear interpolation methods, such as weighted averages, cannot be applied to interpolate intermediate transformations. To address this, we exploit the Lie algebra $\mathfrak{se}(3)$ associated to SE(3). The Lie algebra $\mathfrak{se}(3)$ is a linear 6D vector space corresponding to the tangent space of SE(3) at the identity transformation. Transferring any transformation in SE(3) to a 6D vector element in $\mathfrak{se}(3)$ can be achieved using the logarithm map $\text{Log} : SE(3) \to \mathbb{R}^6$; conversely, the exponential map $\text{Exp} : \mathbb{R}^6 \to SE(3)$ transforms a 6D vector in $\mathfrak{se}(3)$ back to the SE(3) manifold. Therefore, we can first project the SE(3) transformation to $\mathfrak{se}(3)$, perform linear interpolation in this tangent space, and then convert the interpolated vector back to SE(3) to obtain the interpolated transformation. Formally, our transformation interpolation function $\mathcal{F}$ is thus expressed as

$$\mathcal{F}(\sqrt{\bar{\alpha}_t}; \mathbf{H}_0, \mathbb{H}) = \text{Exp}((1 - \sqrt{\bar{\alpha}_t}) \cdot \text{Log}(\mathbb{H}\mathbf{H}_0^{-1}))\mathbf{H}_0, \tag{6}$$

which starts by calculating the relative transformation $\mathbb{H}\mathbf{H}_0^{-1}$ from the optimal transformation $\mathbf{H}_0$ to the identity transformation $\mathbb{H}$, and mapping it to the linear Lie algebra $\mathfrak{se}(3)$ through the logarithmic map. In this linear space, we then scale the vector $\text{Log}(\mathbb{H}\mathbf{H}_0^{-1})$ by the interpolation weight $1 - \sqrt{\bar{\alpha}_t}$, and map this weighted vector back into the corresponding weighted relative transformation using the exponential map. The weighted relative transformation quantifies the offset of the interpolated transformation relative to $\mathbf{H}_0$. Finally, multiplying the optimal transformation $\mathbf{H}_0$ by the computed transformation offset yields the desired interpolated transformation at time step $t$.

**Perturbation Transformation** $\text{Exp}(\gamma\sqrt{1 - \bar{\alpha}_t}\varepsilon)$. Following the standard diffusion formula in Eq. 1, we introduce random noise (*i.e.*, a perturbation transformation) into the interpolated transformation $\mathcal{F}(\sqrt{\bar{\alpha}_t}; \mathbf{H}_0, \mathbb{H})$ at each time step to randomize our SE(3) diffusion process. As indicated in Eq. 1, conventional Euclidean diffusion models typically draw such noise from a Gaussian distribution in Euclidean space. However, formulating a Gaussian distribution on the SE(3) manifold is non-trivial. To tackle this issue, we utilize the Lie algebra once more. Specifically, we randomly sample a 6D noise vector $\varepsilon \in \mathbb{R}^6$ from $\mathcal{N}(\mathbf{0}, \mathbf{I})$ over $\mathbb{R}^6$, which can be interpreted as an element in $\mathfrak{se}(3)$. Then, we scale the noise vector by the factor $\gamma\sqrt{1 - \bar{\alpha}_t}$ as in Eq. 1 to control the magnitude of the perturbation at different time steps. Finally, this scaled noise vector is converted back to SE(3)

using the exponential map to obtain the corresponding perturbation transformation. Please refer to Appendix A for more details about our perturbation transformation.

### 3.2.2 SE(3) Reverse Process

With the diffusion Markov chain $\mathbf{H}_0 \to \mathbf{H}_1 \to \cdots \to \mathbf{H}_T$ generated by the SE(3) diffusion process in Sec. 3.2.1, the objective of the SE(3) reverse process is to train a denoising network $p_\theta(\mathbf{H}_{t-1} \mid \mathcal{X}_t = \mathcal{T}(\mathcal{X}, \mathbf{H}_t), \mathcal{M}))$ to progressively refine the noisy transformation towards the optimal one, thus forming a reverse Markov chain $\mathbf{H}_T \to \mathbf{H}_{T-1} \to \cdots \to \mathbf{H}_0$. In the context of 3D registration, we design our denoising network to predict a probability distribution over the transformation $\mathbf{H}_{t-1}$ given the model point cloud $\mathcal{M}$ and the transformed source point cloud $\mathcal{X}_t = \mathcal{T}(\mathcal{X}, \mathbf{H}_t) = \{\mathbf{R}_t\mathbf{x}_i + \mathbf{t}_t\}$. Here, $\mathbf{R}_t$ and $\mathbf{t}_t$ denote the rotation matrix and translation vector of transformation $\mathbf{H}_t$. To effectively train our denoising network, we derive the following registration-specific variational lower bound of the log likelihood over the training samples as the optimization objective:

$$\mathbb{E}_{\mathcal{X},\mathcal{M},\mathbf{H}_0 \sim p_{data}} \left[ \ln p_\theta(\mathbf{H}_0 \mid \mathcal{X}, \mathcal{M}) \right] \geq \mathbb{E}_{\substack{\mathbf{H}_{1:T} \sim q \\ \mathcal{X},\mathcal{M},\mathbf{H}_0 \sim p_{data}}} \left[ \ln \frac{p_\theta(\mathbf{H}_{0:T} \mid \mathcal{X}, \mathcal{M})}{q(\mathbf{H}_{1:T} \mid \mathbf{H}_0)} \right]$$

$$= \mathbb{E}_{\substack{\mathbf{H}_{1:T} \sim q \\ \mathcal{X},\mathcal{M},\mathbf{H}_0 \sim p_{data}}} \left[ \underbrace{\ln p_\theta(\mathbf{H}_0 \mid \mathcal{X}_1, \mathcal{M})}_{\text{Residual term}} - \underbrace{D_{\mathrm{KL}}(q(\mathbf{H}_T \mid \mathbf{H}_0) \| p(\mathbf{H}_T))}_{\text{Prior matching term}} - \right. \tag{7}$$

$$\left. \sum_{t=2}^{T} \underbrace{D_{\mathrm{KL}}(q(\mathbf{H}_{t-1} \mid \mathbf{H}_t, \mathbf{H}_0) \| p_\theta(\mathbf{H}_{t-1} \mid \mathcal{X}_t, \mathcal{M}))}_{\text{Denoising matching term}} \right],$$

where $p_{data}$ indicates the distribution of the training data, including the pairs of source and model point clouds and their ground-truth rigid transformations. In the following, we elaborate on each loss term of the derived variational lower bound. Please refer to Appendix A for a detailed derivation.

**Denoising Matching Term.** This is the fundamental loss term for training our denoising network. $q(\mathbf{H}_{t-1} \mid \mathbf{H}_t, \mathbf{H}_0)$ represents the posterior distribution of $\mathbf{H}_{t-1}$ conditioned on $\mathbf{H}_0$ and $\mathbf{H}_t$, while $p_\theta(\mathbf{H}_{t-1} \mid \mathcal{X}_t, \mathcal{M})$ denotes the prior distribution over $\mathbf{H}_{t-1}$ predicted by our denoising network. In contrast to the prior distribution, the posterior has access to the optimal transformation $\mathbf{H}_0$, enabling it to infer a more reliable distribution for $\mathbf{H}_{t-1}$ through Bayes' formula. As a result, we can treat this posterior distribution as ground-truth signal to supervise the prior distribution prediction of the denoising network by minimizing their Kullback-Leibler (KL) divergence. Inspired by the Bayesian posterior in Eq. 3, the random transformation $\mathbf{H}_{t-1}^{post}$ following the posterior distribution, $\mathbf{H}_{t-1}^{post} \sim q(\mathbf{H}_{t-1} \mid \mathbf{H}_t, \mathbf{H}_0)$, can be expressed as

$$\mathbf{H}_{t-1}^{post} = \mathrm{Exp}\left( \underbrace{\frac{\sqrt{\bar{\alpha}_{t-1}}\beta_t}{1 - \bar{\alpha}_t}}_{\lambda_0} \mathrm{Log}\,(\mathbf{H}_0) + \underbrace{\frac{\sqrt{\alpha_t}\,(1 - \bar{\alpha}_{t-1})}{1 - \bar{\alpha}_t}}_{\lambda_1} \mathrm{Log}\,(\mathbf{H}_t) + \sqrt{\tilde{\beta}_t}\varepsilon \right), \tag{8}$$

where we employ the logarithm map to convert the SE(3) transformations $\mathbf{H}_0$ and $\mathbf{H}_t$ to their corresponding 6D vector representations in the linear Lie algebra $\mathfrak{se}(3)$. This allows us to perform a linear combination with posterior coefficients $\lambda_0$ and $\lambda_1$, as in Eq. 3. The randomness of $\mathbf{H}_{t-1}^{post}$ arises from the addition of a random variable $\varepsilon$, which follows a Gaussian distribution $\mathcal{N}(\mathbf{0}, \mathbf{I})$ over $\mathbb{R}^6$. The resulting vector is then converted back from $\mathfrak{se}(3)$ to SE(3) using the exponential map, yielding $\mathbf{H}_{t-1}^{post}$. Analogously, the random transformation following the prior distribution, $\mathbf{H}_{t-1}^{prior} \sim p_\theta(\mathbf{H}_{t-1} \mid \mathcal{X}_t, \mathcal{M})$, can be represented as

$$\mathbf{H}_{t-1}^{prior} = \mathrm{Exp}\left( \mathrm{Log}\,(\mu_\theta(\mathcal{X}_t, \mathcal{M})) + \sqrt{\tilde{\beta}_t}\varepsilon \right), \tag{9}$$

where $\mu_\theta(\mathcal{X}_t, \mathcal{M})$ denotes the parameterized mean of our prior distribution, and $\sqrt{\tilde{\beta}_t}\varepsilon$ represents the random term that is identical to that of the posterior. As such, minimizing the KL divergence between the prior and posterior distributions is equivalent to minimizing the error between $\mathrm{Log}(\mu_\theta(\mathcal{X}_t, \mathcal{M}))$ and $\lambda_0 \mathrm{Log}(\mathbf{H}_0) + \lambda_1 \mathrm{Log}(\mathbf{H}_t)$, i.e.,

$$\mathcal{L}_t(\theta) = loss(\lambda_0 \mathrm{Log}(\mathbf{H}_0) + \lambda_1 \mathrm{Log}(\mathbf{H}_t), \mathrm{Log}(\mu_\theta(\mathcal{X}_t, \mathcal{M}))) . \tag{10}$$

The loss function 10 indicates that, at each time step $t$, given the point clouds $\mathcal{X}_t = \mathcal{T}(\mathcal{X}, \mathbf{H}_t)$ and $\mathcal{M}$, the mean $\mu_\theta(\mathcal{X}_t, \mathcal{M})$ needs to be predicted as the transformation $\text{Exp}(\lambda_0 \log(\mathbf{H}_0) + \lambda_1 \log(\mathbf{H}_t))$ rather than the relative transformation $\mathbf{H}_0 \mathbf{H}_t^{-1}$ between $\mathcal{X}_t$ and $\mathcal{M}$. Consequently, existing deep registration models such as [54, 62] may not be directly applicable for parameterizing the mean of our prior distribution. Moreover, designing a specific network that takes $\mathcal{X}_t$ and $\mathcal{M}$ as input but predicts a non-relative transformation also is a non-trivial task. To handle it, we consider that the transformation $\text{Exp}(\lambda_0 \text{Log}(\mathbf{H}_0) + \lambda_1 \log(\mathbf{H}_t))$ can be rewritten as

$$\text{Exp}\left(\lambda_0 \text{Log}(\mathbf{H}_0) + \lambda_1 \text{Log}(\mathbf{H}_t)\right) = \text{Exp}\left(\lambda_0 \text{Log}(\underbrace{(\mathbf{H}_0 \mathbf{H}_t^{-1})}_{\mathbf{H}_{t \to 0}} \mathbf{H}_t) + \lambda_1 \text{Log}(\mathbf{H}_t)\right). \tag{11}$$

This inspires us to reformulate $\mu_\theta(\mathcal{X}_t, \mathcal{M})$ using a surrogate registration model $f_\theta(\mathcal{X}_t, \mathcal{M})$ as

$$\mu_\theta(\mathcal{X}_t, \mathcal{M}) = \text{Exp}\left(\lambda_0 \text{Log}(f_\theta(\mathcal{X}_t, \mathcal{M})\mathbf{H}_t) + \lambda_1 \text{Log}(\mathbf{H}_t)\right). \tag{12}$$

Then, minimizing the loss function 10 is equivalent to optimizing the surrogate registration model $f_\theta(\mathcal{X}_t, \mathcal{M})$ to predict the relative transformation $\mathbf{H}_{t \to 0} = \mathbf{H}_0 \mathbf{H}_t^{-1}$ between the point clouds $\mathcal{X}_t$ and $\mathcal{M}$. Therefore, different deep registration models can potentially serve as surrogate registration models in our denoising network. Finally, we optimize the surrogate registration network by minimizing the $L_1$ distance between the source points transformed using the ground-truth transformation $\mathbf{H}_{t \to 0}$ and the predicted one $\hat{\mathbf{H}}_{t \to 0} = f_\theta(\mathcal{X}_t, \mathcal{M})$. This is written as

$$\mathcal{L}_t(\theta) = loss(f_\theta(\mathcal{X}_t, \mathcal{M}), \mathbf{H}_{t \to 0}) = \frac{1}{N} \sum_i^N \left\| \mathbf{H}_{t \to 0} \begin{bmatrix} \mathbf{x}_i^t \\ 1 \end{bmatrix} - \hat{\mathbf{H}}_{t \to 0} \begin{bmatrix} \mathbf{x}_i^t \\ 1 \end{bmatrix} \right\|_1, \tag{13}$$

where $t \in \{2, ..., T\}$ and $\mathbf{x}_i^t$ indicates the $i$-th point in the transformed source point cloud $\mathcal{X}_t$.

**Residual and Prior Matching Terms.** To maximize the probability $p_\theta(\mathbf{H}_0 \mid \mathcal{X}_1, \mathcal{M})$, the mean $\mu_\theta(\mathcal{X}_1, \mathcal{M}) = \text{Exp}\left(\lambda_0 \text{Log}(f_\theta(\mathcal{X}_1, \mathcal{M})\mathbf{H}_1) + \lambda_1 \text{Log}(\mathbf{H}_1)\right)$ should be optimized to align closely with $\mathbf{H}_0$. As such, the optimization target of $f_\theta(\mathcal{X}_1, \mathcal{M})$ is $\mathbf{H}_0 \mathbf{H}_1^{-1}$, and the loss function can be represented as $\mathcal{L}_1(\theta)$ in 13. In addition, as the prior matching term does not require learning any parameters, it can be regarded as a constant and hence can be omitted.

## 4 Experiments

### 4.1 Experimental Settings

**Implementation Details.** We set the numbers of points in the source and model point clouds to $N = 512$ and $M = 1024$ through random sampling. For the SE(3) diffusion process, we adopt a cosine schedule [35] to determine the diffusion coefficients $\{\beta_t\}$. The number of diffusion steps $T$ is set to 200, and the scaling coefficient $\gamma$ for the perturbation transformation is set to 0.1. For the SE(3) reverse process, the number of reverse steps in the training phase is set to 200, while that in the inference phase is set to 5 to accelerate the inference speed of the diffusion registration. We use the ADAM optimizer [25] with a learning rate of 0.001 to optimize the loss function 13 for 20 epochs with a batch size of 32, and employ PyTorch [39] to implement our framework. All experiments are conducted on a server equipped with an Intel i5 2.2 GHz CPU and one TITAN RTX GPU.

**Evaluation Metrics.** Following [11], we evaluate the model performance by quantifying the rotation and translation errors between the predicted rotation and translation $\hat{\mathbf{R}}$ and $\hat{\mathbf{t}}$, and the ground-truth ones $\mathbf{R}^*$ and $\mathbf{t}^*$. The evaluation metrics are defined as

$$\text{RE}(\hat{\mathbf{R}}) = \arccos \frac{\text{Tr}\left(\hat{\mathbf{R}}^\top \mathbf{R}^*\right) - 1}{2}, \quad \text{TE}(\hat{\mathbf{t}}) = \left\|\hat{\mathbf{t}} - \mathbf{t}^*\right\|_2^2. \tag{14}$$

As in [12, 11], we summarize these errors via mean average precision (mAP) under varying thresholds.

### 4.2 Comparison with Existing Methods

**Evaluation on TUD-L.** We first evaluate our approach on the TUD-L dataset [20], a real-world dataset comprising three household objects. The compared methods encompass four representative

| Models | TUD-L | | | | LINEMOD | | | | Occluded-LINEMOD | | | |
|---|---|---|---|---|---|---|---|---|---|---|---|---|
| | 5° | 10° | 1cm | 2cm | 5° | 10° | 1cm | 2cm | 5° | 10° | 1cm | 2cm |
| ICP [4] | 0.02 | 0.02 | 0.01 | 0.14 | 0.00 | 0.01 | 0.04 | 0.27 | 0.01 | 0.01 | 0.07 | 0.36 |
| FGR [66] | 0.00 | 0.01 | 0.04 | 0.25 | 0.00 | 0.00 | 0.05 | 0.31 | 0.00 | 0.00 | 0.08 | 0.43 |
| TEASER [59] | 0.13 | 0.17 | 0.03 | 0.22 | 0.01 | 0.03 | 0.03 | 0.21 | 0.01 | 0.02 | 0.04 | 0.26 |
| S4PCS [34] | 0.30 | 0.50 | 0.05 | 0.40 | 0.02 | 0.09 | 0.04 | 0.31 | 0.01 | 0.03 | 0.06 | 0.31 |
| IDAM [28] | 0.03 | 0.05 | 0.02 | 0.08 | 0.00 | 0.01 | 0.03 | 0.16 | 0.00 | 0.02 | 0.07 | 0.26 |
| FMR [22] | 0.02 | 0.09 | 0.02 | 0.06 | 0.00 | 0.01 | 0.07 | 0.17 | 0.00 | 0.00 | 0.09 | 0.18 |
| RGM [15] | 0.00 | 0.00 | 0.02 | 0.03 | 0.00 | 0.00 | 0.07 | 0.15 | 0.00 | 0.00 | 0.09 | 0.22 |
| RIENet [44] | 0.00 | 0.00 | 0.06 | 0.11 | – | – | – | – | – | – | – | – |
| MN-IDAM [11] | 0.36 | 0.46 | 0.23 | 0.47 | 0.01 | 0.07 | 0.13 | 0.38 | 0.02 | 0.08 | 0.15 | 0.44 |
| MN-DCP [11] | 0.70 | 0.81 | 0.71 | 0.86 | 0.10 | 0.27 | 0.26 | 0.60 | 0.07 | _0.19_ | 0.24 | **0.57** |
| DCP [54] | 0.23 | 0.62 | 0.04 | 0.26 | 0.06 | 0.22 | 0.11 | 0.27 | 0.03 | 0.12 | 0.11 | 0.27 |
| **Diff-DCP** | 0.65 | 0.85 | 0.73 | _0.94_ | **0.22** | **0.51** | **0.65** | **0.82** | _0.10_ | **0.29** | **0.38** | **0.57** |
| _Improvement ↑_ | _0.42_ | _0.23_ | _0.69_ | _0.68_ | _0.16_ | _0.29_ | _0.54_ | _0.55_ | _0.07_ | _0.17_ | _0.27_ | _0.30_ |
| RPMNet [62] | _0.73_ | _0.97_ | _0.89_ | _0.94_ | 0.05 | 0.18 | 0.22 | 0.45 | 0.03 | 0.13 | 0.22 | 0.40 |
| **Diff-RPMNet** | **0.90** | **0.98** | **0.98** | **0.99** | _0.18_ | _0.47_ | _0.51_ | _0.72_ | **0.12** | **0.29** | _0.36_ | _0.52_ |
| _Improvement ↑_ | _0.17_ | _0.01_ | _0.09_ | _0.05_ | _0.13_ | _0.29_ | _0.29_ | _0.27_ | _0.09_ | _0.16_ | _0.14_ | _0.12_ |

Table 1: Quantitative comparisons on TUD-L [20], LINEMOD [18], and Occluded-LINEMOD [6].

traditional approaches: ICP [4], FGR [66], TEASER [60], and S4PCS [34], alongside eight state-of-the-art learning-based deep registration models: DCP [54], IDAM [28], FMR [22], RPMNet [62], RGM [15], RIENet [44], MN-IDAM [11] and MN-DCP [11]. Although Sec. 3.2.2 revealed that many deep registration models are theoretically applicable to establish our denoising network, the empirical results in Table 1 indicate that some models such as IDAM, FMR, RGM, and RIENet fail to produce meaningful results in real-world challenges, suggesting their limited capability in learning a high-quality denoising network through loss function optimization (see Eq.13). Consequently, we opt to employ DCP and RPMNet, which demonstrate promising performance, to establish our denoising network and generate their corresponding diffusion variants: **Diff-DCP** and **Diff-RPMNet**. Table 1 (left block) shows that the real-world challenges posed by TUD-L lead to limited performance exhibited by the compared traditional and deep methods. In contrast, Diff-RPMNet achieves the highest registration accuracy across all rotation and translation criteria. Furthermore, both Diff-DCP and Diff-RPMNet outperform their respective baselines, DCP and RPMNet, by a substantial margin, particularly impressive for the 5°@mAP (42%↑) and 5cm@mAP (69%↑) improvements achieved by Diff-DCP. Such superior performance can be primarily attributed to two factors: **(i)** The Bayesian posterior in Eq.8 effectively orchestrates the pose change of the source point cloud at each reverse step, mitigating premature convergence to local optima; **(ii)** The diffusion process generates a diverse range of poses for the source point cloud, facilitating more comprehensive model training. Table 2 (top block) also strongly confirms these two factors: The baseline DCP model, trained using samples generated by our diffusion process (*DiffAug*), presents a remarkable precision improvement (factor (ii)). Furthermore, when equipped with the reverse process (*Rev.*), its performance is further enhanced (factor (i)). Some qualitative results are provided in Fig.2. Please see Appendix B for additional qualitative results.

**Evaluation on LINEMOD and Occluded-LINEMOD.** We further assess the performance of our method on two widely-used real-world 6D object pose estimation datasets: LINEMOD [18] and Occluded-LINEMOD [6]. The former dataset encompasses 15 texture-less household objects situated in cluttered scenes, while the latter is a subset comprising 8 texture-less objects with varying degrees of occlusion. As shown in Table 1 (middle and right blocks), our Diff-DCP and Diff-RPMNet consistently outperform the compared methods across nearly all rotation and translation mAP criteria, yielding remarkable improvements over their respective baselines, DCP and RPMNet.

## 5 Ablation Studies and Analysis

**Diffusion Process. (1)** We first test the performance variations under different noise schedules, namely the linear schedule [19] and cosine schedule [35], using Diff-DCP for our ablation study. As shown in Table 2 (second block), the cosine schedule tends to yield higher precision on the more challenging LINEMOD and Occluded-LINEMOD datasets, whereas the linear schedule performs better on the comparatively easier TUD-L dataset. We attribute this discrepancy to the fact that the transformation denoising during reverse process in challenging datasets (*e.g.*, LINEMOD) necessitates

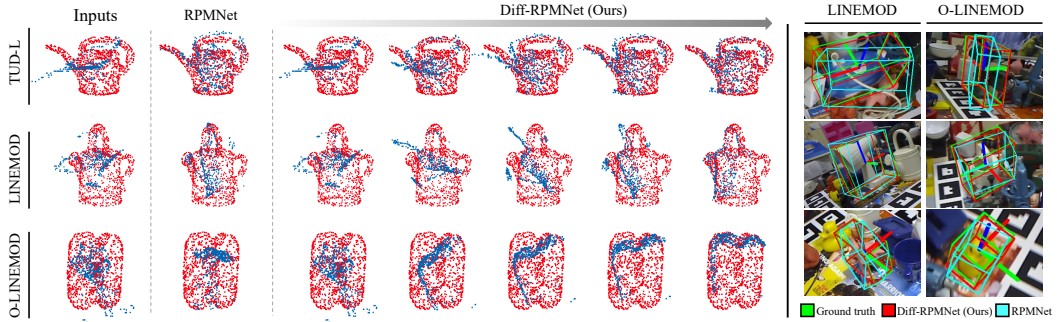

Figure 2: Qualitative comparisons on TUD-L [20], LINEMOD [18], and Occluded-LINEMOD [6].

| Models | TUD-L | | | | LINEMOD | | | | Occluded-LINEMOD | | | |
|---|---|---|---|---|---|---|---|---|---|---|---|---|
| | 5° | 10° | 1cm | 2cm | 5° | 10° | 1cm | 2cm | 5° | 10° | 1cm | 2cm |
| DCP | 0.23 | 0.62 | 0.04 | 0.26 | 0.06 | 0.22 | 0.11 | 0.27 | 0.03 | 0.12 | 0.11 | 0.27 |
| DCP+DiffAug | 0.48 | 0.82 | 0.56 | 0.88 | 0.16 | 0.42 | 0.52 | 0.71 | 0.07 | 0.24 | 0.33 | 0.51 |
| DCP+DiffAug+Rev.* | **0.65** | **0.85** | **0.73** | **0.94** | **0.22** | **0.51** | **0.65** | **0.82** | **0.10** | **0.29** | **0.38** | **0.57** |
| Linear schedule | **0.78** | **0.93** | **0.93** | **0.97** | 0.16 | 0.44 | 0.56 | 0.75 | **0.10** | 0.26 | 0.35 | 0.54 |
| Cosine schedule* | 0.65 | 0.85 | 0.73 | 0.94 | **0.22** | **0.51** | **0.65** | **0.82** | **0.10** | **0.29** | **0.38** | **0.57** |
| Random infer. | 0.63 | **0.86** | 0.68 | 0.92 | **0.22** | 0.50 | 0.62 | 0.80 | **0.11** | **0.29** | **0.38** | 0.55 |
| Deterministic infer.* | **0.65** | 0.85 | **0.73** | **0.94** | **0.22** | **0.51** | **0.65** | **0.82** | 0.10 | **0.29** | **0.38** | **0.57** |
| Train. steps $T = 50$ | 0.61 | 0.83 | 0.65 | 0.87 | 0.19 | 0.47 | 0.54 | 0.73 | 0.10 | 0.27 | 0.35 | 0.55 |
| Train. steps $T = 100$ | **0.70** | **0.92** | **0.87** | **0.95** | 0.18 | 0.46 | 0.61 | 0.80 | **0.12** | 0.28 | **0.42** | **0.57** |
| Train. steps $T = 200$* | 0.65 | 0.85 | 0.73 | 0.94 | **0.22** | **0.51** | **0.65** | **0.82** | 0.10 | **0.29** | 0.38 | **0.57** |

Table 2: Ablation studies on TUD-L [20], LINEMOD [18], and Occluded-LINEMOD [6].

the rich sample diversity offered by the cosine schedule. In contrast, an abundance of sample diversity on TUD-L would lead to reduced sample efficiency (many generated pose samples of the source point cloud are useless), thus resulting in a degradation of model performance. **(2)** Additionally, we investigate the registration precision of Diff-DCP across different training steps. Table 2 (bottom block) illustrates that, compared to a small number of steps, such as $T = 50$, employing a larger number of diffusion steps, such as 100 and 200, leads to higher precision. This observation stems from the fact that a greater number of diffusion steps significantly enhances sample diversity. Notably, Diff-DCP with $T = 200$ cannot achieve the best performance on TUD-L. This further validates our claim that too rich sample diversity on the easier TUD-L does not bring higher performance due to low sample efficiency.

**Reverse Process. (1)** We evaluate our model using two inference strategies: deterministic inference and random inference. In deterministic inference, the denoised transformation at each time step is equal to the predicted mean from Eq. 12, while in random inference, the mean is slightly perturbed by the sampled noise as described in Eq. 9. As shown in Table 2 (third block), deterministic inference without noise tends to exhibit greater stability and achieve lower estimation errors. **(2)** To verify the effectiveness of the Bayesian posterior (Eq. 8) in guiding the pose change of the source point cloud at each reverse step, we plot the mean error changes of the estimated transformations at different denoising time steps. Fig. 3 confirms that, scheduled by the Bayesian posterior, the denoised transformation gradually approaches to the optimal transformation. Furthermore, the table in Fig. 3 indicates that employing a larger number of reverse steps gradually increases the inference time. Therefore, in our implementation, we set the inference step to 5 to improve the registration efficiency.

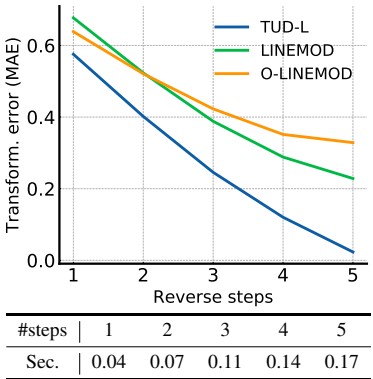

| #steps | 1 | 2 | 3 | 4 | 5 |
|---|---|---|---|---|---|
| Sec. | 0.04 | 0.07 | 0.11 | 0.14 | 0.17 |

Figure 3: Variations in prediction error and runtime for different time steps during the reverse process.

## 6 Conclusion

In this paper, we have proposed a novel and effective SE(3) diffusion model-based point cloud registration framework for robust 6D object pose estimation in real-world scenarios. The framework

formulates point cloud registration as a denoising diffusion process, enabling progressive refinement of the pose of the source point cloud for precise alignment with the model point cloud. To facilitate the diffusion and reverse processes over the SE(3) manifold, we have introduced the Lie algebra $\mathfrak{se}(3)$ associated with SE(3) to constrain the transformation transitions. Furthermore, we have derived a registration-specific variational lower bound to effectively optimize our denoising network. By reformulating our denoising network with a surrogate registration model, different deep registration networks can theoretically be employed within our approach. Our extensive experiments on challenging real-world datasets have validated the effectiveness of our framework. We discuss the broader impact, limitations, and future work in Appendix C.

## 7 Acknowledgments

The authors would like to thank the editor and the anonymous reviewers for their critical and constructive comments and suggestions. This work was supported by the National Science Fund of China under (Grant Nos. U1713208, 62361166670, 62276144) and the Swiss Innovation Agency (Innosuisse).

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
