# Supplementary Material for "SE(3) Diffusion Model-based Point Cloud Registration for Robust 6D Object Pose Estimation"

**Haobo Jiang**[1], **Mathieu Salzmann**[2,3*], **Zheng Dang**[2], **Jin Xie**[1*], and **Jian Yang**[1*]

[1]PCA Lab, Nanjing University of Science and Technology, China
[2]CVLab, EPFL, Switzerland
[3]ClearSpace, Switzerland
{jiang.hao.bo, csjxie, csjyang}@njust.edu.cn
{zheng.dang, mathieu.salzmann}@epfl.ch

## A   More Details about SE(3) Diffusion Model for Point Cloud Registration

### A.1   Proof of Variational Lower Bound

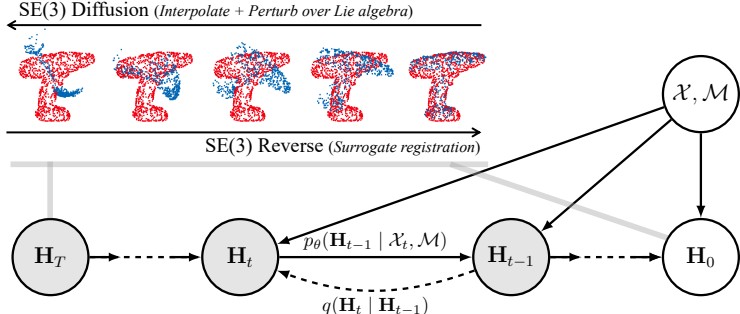

Figure 1: Probabilistic graphical model of our SE(3) diffusion model-based registration framework.

Based on the established probabilistic graphical model in Fig. 1, the variational lower bound on our SE(3) diffusion model for point cloud registration can be derived as below. Specifically, we first inject a set of latent transformation variables $\{\mathbf{H}_1, \mathbf{H}_2, ..., \mathbf{H}_T\}$ into the log likelihood of the training data $\{\mathcal{X}, \mathcal{M}, \mathbf{H}_0\} \sim p_{data}$ and the initial variational lower bound can be derived as:

$$\ln p_\theta(\mathbf{H}_0 \mid \mathcal{X}, \mathcal{M}) = \ln \int_{\mathbf{H}_{1:T}} p_\theta(\mathbf{H}_{0:T} \mid \mathcal{X}, \mathcal{M}) d_{\mathbf{H}_{1:T}}$$

$$= \ln \int_{\mathbf{H}_{1:T}} q(\mathbf{H}_{1:T} \mid \mathbf{H}_0) \frac{p_\theta(\mathbf{H}_{0:T} \mid \mathcal{X}, \mathcal{M})}{q(\mathbf{H}_{1:T} \mid \mathbf{H}_0)} d_{\mathbf{H}_{1:T}} = \ln \mathbb{E}_{\mathbf{H}_{1:T} \sim q} \left[ \frac{p_\theta(\mathbf{H}_{0:T} \mid \mathcal{X}, \mathcal{M})}{q(\mathbf{H}_{1:T} \mid \mathbf{H}_0)} \right] \quad (1)$$

$$\overset{(1)}{\geq} \mathbb{E}_{\mathbf{H}_{1:T} \sim q} \left[ \ln \frac{p_\theta(\mathbf{H}_{0:T} \mid \mathcal{X}, \mathcal{M})}{q(\mathbf{H}_{1:T} \mid \mathbf{H}_0)} \right] \overset{(2)}{\cong} \mathbb{E}_{\mathbf{H}_{1:T} \sim q} \left[ \ln \frac{p(\mathbf{H}_T) \cdot p_\theta(\mathbf{H}_{0:T-1} \mid \mathcal{T}(\mathcal{X}, \mathbf{H}_T), \mathcal{M})}{q(\mathbf{H}_{1:T} \mid \mathbf{H}_0)} \right]$$

where the inequality (1) is based on the Jensen's inequality and the step (2) is based on the chain rule in probability theory; $p(\mathbf{H}_T)$ denotes the prior transformation distribution. Next, based on the defined

---

*Corresponding authors.

Haobo Jiang, Jin Xie, and Jian Yang are with PCA Lab, Key Lab of Intelligent Perception and Systems for High-Dimensional Information of Ministry of Education, and Jiangsu Key Lab of Image and Video Understanding for Social Security, School of Computer Science and Engineering, Nanjing University of Science and Technology, China.

37th Conference on Neural Information Processing Systems (NeurIPS 2023).

conditional dependencies of random variables in our probabilistic graphical model (Fig. 1), the chain rule is further used to factorize the posterior distribution $q(\mathbf{H}_{1:T} \mid \mathbf{H}_0)$ and the prior distribution $p_\theta(\mathbf{H}_{0:T} \mid \mathcal{X}, \mathcal{M})$ as below:

$$q(\mathbf{H}_{1:T} \mid \mathbf{H}_0) = q(\mathbf{H}_1 \mid \mathbf{H}_0) \cdot q(\mathbf{H}_{2:T} \mid \mathbf{H}_1) = q(\mathbf{H}_1 \mid \mathbf{H}_0) \cdot q(\mathbf{H}_2 \mid \mathbf{H}_1) \cdot q(\mathbf{H}_{3:T} \mid \mathbf{H}_2) = ... = \prod_{t=1}^{T} q(\mathbf{H}_t \mid \mathbf{H}_{t-1})$$

(2)

$$
\begin{aligned}
&p_\theta(\mathbf{H}_{0:T-1} \mid \mathcal{T}(\mathcal{X}, \mathbf{H}_T), \mathcal{M}) \\
=&p_\theta(\mathbf{H}_{T-1} \mid \mathcal{T}(\mathcal{X}, \mathbf{H}_T), \mathcal{M}) \cdot p_\theta(\mathbf{H}_{0:T-2} \mid \mathcal{T}(\mathcal{X}, \mathbf{H}_{T-1}), \mathcal{M}) \\
=&p_\theta(\mathbf{H}_{T-1} \mid \mathcal{T}(\mathcal{X}, \mathbf{H}_T), \mathcal{M}) \cdot p_\theta(\mathbf{H}_{T-2} \mid \mathcal{T}(\mathcal{X}, \mathbf{H}_{T-1}), \mathcal{M}) \cdot p_\theta(\mathbf{H}_{0:T-3} \mid \mathcal{T}(\mathcal{X}, \mathbf{H}_{T-2}), \mathcal{M}) \\
=&\prod_{t=1}^{T} p_\theta(\mathbf{H}_{t-1} \mid \mathcal{T}(\mathcal{X}, \mathbf{H}_t), \mathcal{M})
\end{aligned}
$$

(3)

By inserting the factorized posterior (Eq. 3) and prior (Eq. 2) distributions into Eq. 1, we can derive the detailed variational lower bound as below:

$$
\begin{aligned}
&\ln p_\theta(\mathbf{H}_0 \mid \mathcal{X}, \mathcal{M}) \\
\geq&\mathbb{E}_{\mathbf{H}_{1:T} \sim q} \left[ \ln \frac{p(\mathbf{H}_T) \cdot p_\theta(\mathbf{H}_{0:T-1} \mid \mathcal{T}(\mathcal{X}, \mathbf{H}_T), \mathcal{M})}{q(\mathbf{H}_{1:T} \mid \mathbf{H}_0)} \right] \\
=&\mathbb{E}_{\mathbf{H}_{1:T} \sim q} \left[ \ln \frac{p(\mathbf{H}_T) \prod_{t=1}^{T} p_\theta(\mathbf{H}_{t-1} \mid \mathcal{T}(\mathcal{X}, \mathbf{H}_t), \mathcal{M})}{\prod_{t=1}^{T} q(\mathbf{H}_t \mid \mathbf{H}_{t-1})} \right] \\
=&\mathbb{E}_{\mathbf{H}_{1:T} \sim q} \left[ \ln p(\mathbf{H}_T) + \sum_{t=1}^{T} \ln \frac{p_\theta(\mathbf{H}_{t-1} \mid \mathcal{T}(\mathcal{X}, \mathbf{H}_t), \mathcal{M})}{q(\mathbf{H}_t \mid \mathbf{H}_{t-1})} \right]
\end{aligned}
$$

(4)

Particularly, using Bayes' formula, $q(\mathbf{H}_t \mid \mathbf{H}_{t-1})$ can be deformed as below:

$$
\begin{aligned}
q(\mathbf{H}_t \mid \mathbf{H}_{t-1}) &= q(\mathbf{H}_t \mid \mathbf{H}_{t-1}, \mathbf{H}_0) = \frac{q(\mathbf{H}_t, \mathbf{H}_{t-1}, \mathbf{H}_0)}{q(\mathbf{H}_{t-1}, \mathbf{H}_0)} \\
&= \frac{q(\mathbf{H}_{t-1} \mid \mathbf{H}_t, \mathbf{H}_0) q(\mathbf{H}_t \mid \mathbf{H}_0) q(\mathbf{H}_0)}{q(\mathbf{H}_{t-1} \mid \mathbf{H}_0) q(\mathbf{H}_0)} = \frac{q(\mathbf{H}_{t-1} \mid \mathbf{H}_t, \mathbf{H}_0) q(\mathbf{H}_t \mid \mathbf{H}_0)}{q(\mathbf{H}_{t-1} \mid \mathbf{H}_0)}.
\end{aligned}
$$

(5)

By inserting Eq. 5 into the variational lower bound 4, we can further rewrite the variational lower bound as below:

$$
\begin{aligned}
&\ln p_\theta(\mathbf{H}_0 \mid \mathcal{X}, \mathcal{M}) \\
\geq&\mathbb{E}_{\mathbf{H}_{1:T} \sim q} \left[ \ln p(\mathbf{H}_T) + \sum_{t=1}^{T} \ln \frac{p_\theta(\mathbf{H}_{t-1} \mid \mathcal{T}(\mathcal{X}, \mathbf{H}_t), \mathcal{M})}{q(\mathbf{H}_t \mid \mathbf{H}_{t-1})} \right] \\
=&\mathbb{E}_{\mathbf{H}_{1:T} \sim q} \left[ \ln p(\mathbf{H}_T) + \sum_{t=1}^{T} \ln \frac{p_\theta(\mathbf{H}_{t-1} \mid \mathcal{T}(\mathcal{X}, \mathbf{H}_t), \mathcal{M})}{\frac{q(\mathbf{H}_{t-1} \mid \mathbf{H}_t, \mathbf{H}_0) q(\mathbf{H}_t \mid \mathbf{H}_0)}{q(\mathbf{H}_{t-1} \mid \mathbf{H}_0)}} \right] \\
=&\mathbb{E}_{\mathbf{H}_{1:T} \sim q} \left[ \ln p(\mathbf{H}_T) + \sum_{t=1}^{T} \ln \frac{p_\theta(\mathbf{H}_{t-1} \mid \mathcal{T}(\mathcal{X}, \mathbf{H}_t), \mathcal{M}) q(\mathbf{H}_{t-1} \mid \mathbf{H}_0)}{q(\mathbf{H}_{t-1} \mid \mathbf{H}_t, \mathbf{H}_0) q(\mathbf{H}_t \mid \mathbf{H}_0)} \right] \\
=&\mathbb{E}_{\mathbf{H}_{1:T} \sim q} \left[ \ln p(\mathbf{H}_T) + \sum_{t=2}^{T} \ln \frac{p_\theta(\mathbf{H}_{t-1} \mid \mathcal{T}(\mathcal{X}, \mathbf{H}_t), \mathcal{M}) q(\mathbf{H}_{t-1} \mid \mathbf{H}_0)}{q(\mathbf{H}_{t-1} \mid \mathbf{H}_t, \mathbf{H}_0) q(\mathbf{H}_t \mid \mathbf{H}_0)} + \ln \frac{p_\theta(\mathbf{H}_0 \mid \mathcal{T}(\mathcal{X}, \mathbf{H}_1), \mathcal{M}) q(\mathbf{H}_0 \mid \mathbf{H}_0)}{q(\mathbf{H}_0 \mid \mathbf{H}_1, \mathbf{H}_0) q(\mathbf{H}_1 \mid \mathbf{H}_0)} \right] \\
=&\mathbb{E}_{\mathbf{H}_{1:T} \sim q} \left[ \ln p(\mathbf{H}_T) + \sum_{t=2}^{T} \ln \frac{p_\theta(\mathbf{H}_{t-1} \mid \mathcal{T}(\mathcal{X}, \mathbf{H}_t), \mathcal{M}) q(\mathbf{H}_{t-1} \mid \mathbf{H}_0)}{q(\mathbf{H}_{t-1} \mid \mathbf{H}_t, \mathbf{H}_0) q(\mathbf{H}_t \mid \mathbf{H}_0)} + \ln \frac{p_\theta(\mathbf{H}_0 \mid \mathcal{T}(\mathcal{X}, \mathbf{H}_1), \mathcal{M})}{q(\mathbf{H}_1 \mid \mathbf{H}_0)} \right] \\
=&\mathbb{E}_{\mathbf{H}_{1:T} \sim q} \left[ \ln \frac{p(\mathbf{H}_T)}{q(\mathbf{H}_T \mid \mathbf{H}_0)} + \sum_{t=2}^{T} \ln \frac{p_\theta(\mathbf{H}_{t-1} \mid \mathcal{T}(\mathcal{X}, \mathbf{H}_t), \mathcal{M})}{q(\mathbf{H}_{t-1} \mid \mathbf{H}_t, \mathbf{H}_0)} + \ln p_\theta(\mathbf{H}_0 \mid \mathcal{T}(\mathcal{X}, \mathbf{H}_1), \mathcal{M}) \right] \\
=&\mathbb{E}_q \left[ \ln p_\theta(\mathbf{H}_0 \mid \mathcal{T}(\mathcal{X}, \mathbf{H}_1), \mathcal{M}) - \mathrm{D}_{\mathrm{KL}}(q(\mathbf{H}_T \mid \mathbf{H}_0) || p(\mathbf{H}_T)) - \sum_{t=2}^{T} \mathrm{D}_{\mathrm{KL}}(q(\mathbf{H}_{t-1} \mid \mathbf{H}_t, \mathbf{H}_0) || p_\theta(\mathbf{H}_{t-1} \mid \mathcal{T}(\mathcal{X}, \mathbf{H}_t), \mathcal{M})) \right]
\end{aligned}
$$

(6)

Finally, considering that source and model point clouds and their ground-truth transformations follow the distribution of training data $p_{data}$, the variational lower bound of the log likelihood over the whole

training data can be expressed as:

$$\mathbb{E}_{\mathcal{X},\mathcal{M},\mathbf{H}_0 \sim p_{data}} \left[ \ln p_\theta(\mathbf{H}_0 \mid \mathcal{X},\mathcal{M}) \right]$$

$$\geq \mathbb{E}_{\substack{\mathbf{H}_{1:T} \sim q \\ \mathcal{X},\mathcal{M},\mathbf{H}_0 \sim p_{data}}} \left[ \ln p_\theta(\mathbf{H}_0 \mid \mathcal{X}_1,\mathcal{M}) - \mathrm{D}_{\mathrm{KL}}(q(\mathbf{H}_T \mid \mathbf{H}_0)||p(\mathbf{H}_T)) - \right.$$

$$\left. \sum_{t=2}^{T} \mathrm{D}_{\mathrm{KL}}(q(\mathbf{H}_{t-1} \mid \mathbf{H}_t, \mathbf{H}_0)||p_\theta(\mathbf{H}_{t-1} \mid \mathcal{X}_t,\mathcal{M})) \right]. \tag{7}$$

## A.2 Details about Perturbation Transformation $\mathrm{Exp}(\gamma\sqrt{1-\bar{\alpha}_t}\varepsilon)$.

As demonstrated in our main paper, we utilize the Lie algebra for randomly sampling the desired perturbation transformation to randomize our SE(3) diffusion process. Specifically, we randomly sample a 6D noise vector $\varepsilon = \begin{bmatrix} \varepsilon_r \\ \varepsilon_t \end{bmatrix} \in \mathbb{R}^6$ from $\mathcal{N}(\mathbf{0},\mathbf{I})$ over $\mathbb{R}^6$, which can be interpreted as an element in $\mathfrak{se}(3)$. Then, we scale the noise vector by the factor $\gamma\sqrt{1-\bar{\alpha}_t}$ to control the magnitude of the perturbation at different time steps:

$$\tilde{\varepsilon} = \gamma\sqrt{1-\bar{\alpha}_t}\varepsilon = \begin{bmatrix} \gamma\sqrt{1-\bar{\alpha}_t}\varepsilon_r \\ \gamma\sqrt{1-\bar{\alpha}_t}\varepsilon_t \end{bmatrix} = \begin{bmatrix} \tilde{\varepsilon}_r \\ \tilde{\varepsilon}_t \end{bmatrix} \in \mathbb{R}^6. \tag{8}$$

Its homogeneous transformation matrix in SE(3) can be written as:

$$\mathrm{Exp}(\tilde{\varepsilon} = \begin{bmatrix} \tilde{\varepsilon}_r \\ \tilde{\varepsilon}_t \end{bmatrix}) = e^{\tilde{\varepsilon}^\wedge} \triangleq \begin{bmatrix} \mathbf{R}(\tilde{\varepsilon}_r) & \mathbf{J}(\tilde{\varepsilon}_r)\tilde{\varepsilon}_t \\ \mathbf{0}^\top & 1 \end{bmatrix} \in \mathbb{R}^{4\times4}, \tag{9}$$

where the rotation matrix $\mathbf{R}(\tilde{\varepsilon}_r)$ and the left-jacobian matrix $\mathbf{J}(\tilde{\varepsilon}_r)$ can be calculated as:

$$\mathbf{R}(\tilde{\varepsilon}_r) = \cos\theta\mathbf{I}_{3\times3} + (1-\cos\theta)\mathbf{u}\mathbf{u}^\top + \sin\theta\mathbf{u}^\wedge,$$
$$\mathbf{J}(\tilde{\varepsilon}_r) = \frac{\sin\theta}{\theta}\mathbf{I}_{3\times3} + \frac{1-\cos\theta}{\theta}\mathbf{u}^\wedge + \frac{\theta-\sin\theta}{\theta}\mathbf{u}\mathbf{u}^\top. \tag{10}$$

Here, the rotation angle $\theta = \|\tilde{\varepsilon}_r\|_2$ and the unit-length direction of the axis of rotation $\mathbf{u} = \tilde{\varepsilon}_r/\theta$.

## B More Visualization Results

In Fig. 2, we provide more visualization results about the reverse process of Diff-RPMNet on TUD-L [5], LINEMOD [4], and Occluded-LINEMOD [2] benchmark datasets, respectively. It can be observed that guided by the Bayesian posterior, the reverse process can progressively refine the pose of the source point cloud to align with the model point cloud precisely.

## C Discussions about Broader Impacts, Limitations and Future Work

**Broader Impacts.** In this section, we clarify the social and academic impacts of our diffusion registration model as follows:

**(1)** For social impact, our diffusion model-based point cloud registration framework has the potential to significantly enhance the accuracy of pure point cloud-based 6D object pose estimation. Consequently, it can make noteworthy contributions to various computer vision applications, including robotics, augmented reality, and autonomous systems, particularly in challenging scenarios such as those involving low-quality lighting conditions. To our best knowledge, we have not identified some evident negative social impacts, and we believe that by implementing rigorous and secure regulations, potential negative impacts can be mitigated.

**(2)** For academic impact, our paper represents the pioneering effort in integrating the diffusion probabilistic model into the SE(3) registration task to achieve more robust 6D object pose estimation. This innovative registration framework exhibits promising registration performance. We hold the conviction that this new framework will inspire the 3D registration community, driving the development of more powerful diffusion model-based registration pipelines. Furthermore, within the domain of

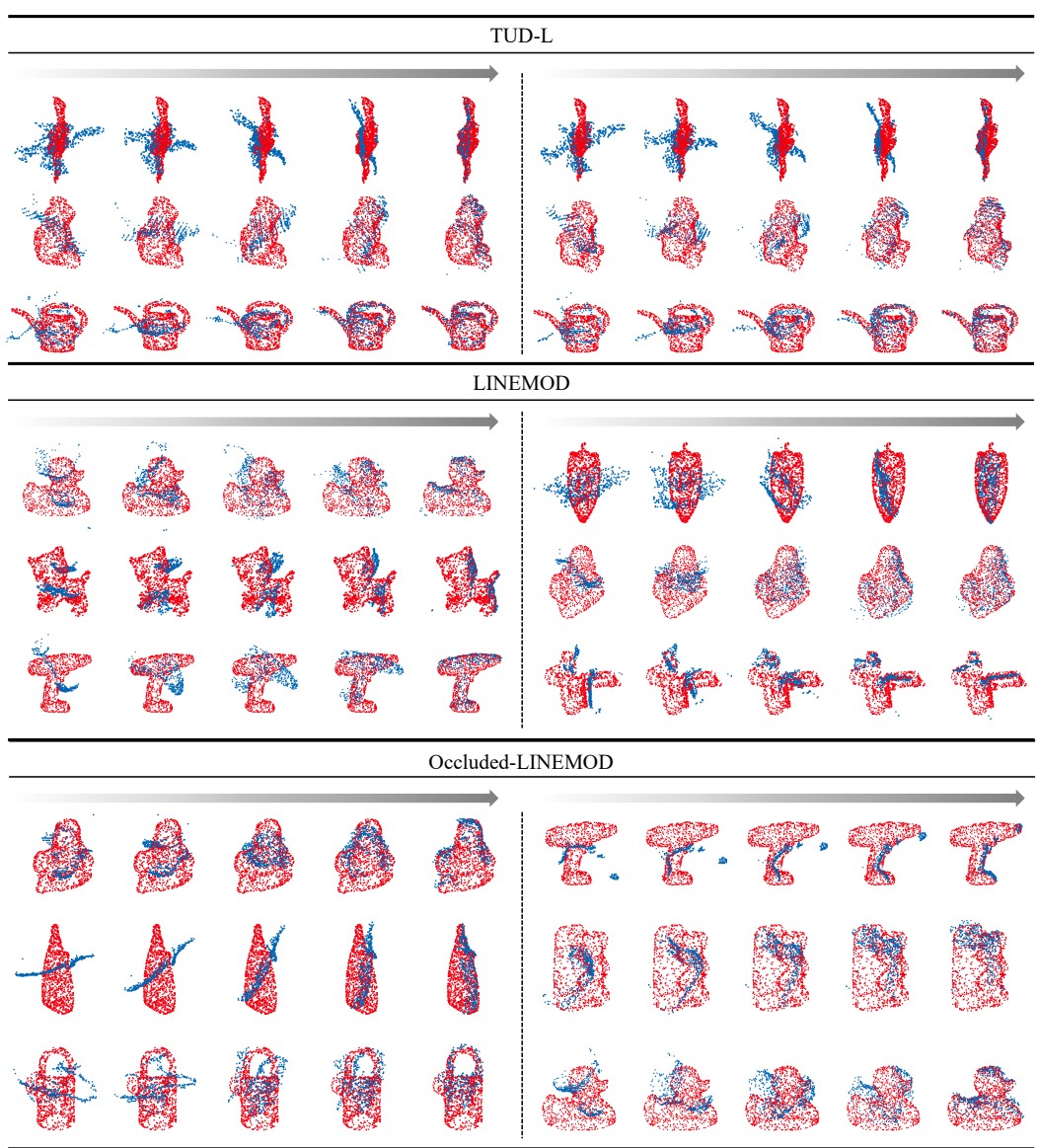

Figure 2: Visualization results of reverse process of Diff-RPMNet on TUD-L [5], LINEMOD [4], and Occluded-LINEMOD [2] benchmark datasets.

pure point cloud-based 6D object pose estimation, we believe our research not only inspires but also encourages the exploration of more correlated pose estimation models. To the best of our knowledge, no apparent negative impact on the academic field has been observed.

**Limitations.** Despite the promising results of our SE(3) diffusion registration model for 6D object pose estimation, several limitations should be acknowledged:

**(1)** Limited inference speed. At each reverse step, our diffusion registration model needs to employ the surrogate registration for computing the Bayesian posterior as the guidance of the pose refinement of the source point cloud. Consequently, as the number of reverse steps increases, the inference time of our model grows linearly, resulting in limited inference efficiency.

**(2)** Similar to [3], our work also primarily focuses on enhancing the robustness of object-level point cloud registration in real-world scenarios. However, it does not address the identification of the source (object) point cloud directly. Instead, we rely on the mask maps provided by the datasets to extract

the source point cloud from the depth maps and leave the point cloud segmentation that segments the source (object) point cloud for future work. It is important to note that despite the usage of the mask map, the resulting source point clouds are still partial due to partial scanning and may contain outliers, particularly along the object's boundaries.

**Future Work.** Those limitations listed above would promote our future research as follows:

**(1)** To improve the inference speed, we can put our reverse process into a compact feature space for reverse inference rather than in the original point cloud space. Specifically, we can employ a 3D encoder (such as PointNet++) to extract global features of the source and model point clouds, serving as contextual information. With this context, we develop a denoising network to directly regress the Bayesian posterior in the feature space rather than relying on time-consuming registration operations in the original point cloud space, thereby significantly reducing the inference time.

**(2)** In our forthcoming work, we intend to integrate point cloud registration and point cloud segmentation into a comprehensive framework. As such, we can first learn to extract the source (object) point cloud from the whole point cloud and then employ our diffusion registration model for estimating the relative transformation as the 6D pose estimation. In addition, we will explore the possibility of designing a diffusion model-guided point cloud segmentation framework to enhance the robustness of segmentation.