# OpenReview forum: "SE(3) Diffusion Model-based Point Cloud Registration for Robust 6D Object Pose Estimation"
_NeurIPS.cc/2023/Conference — NeurIPS 2023 poster_

### Official Review · Reviewer_9j1P · 2023-06-25

**Soundness:** 3 good
**Presentation:** 3 good
**Contribution:** 3 good
**Rating:** 5
**Confidence:** 4

**Summary:**

The paper tackles model-based 6D object pose estimation using SE(3) diffusion model-based point cloud registration. Point cloud registration is trained with diffusion and denoising processes on SE(3), which gradually perturb the optimal pose and learn a denoising network to refine the noisy transformation progressively. The SE(3) optimization objective is derived from a 3D registration-specific variational lower bound and the denoising network is trained with a surrogate registration model. The perturbation and interpolation on SE(3) are performed with Lie algebra. Experiments on TUD-L, LINEMOD, and Occluded-LINEMOD datasets show the validity of the method.

**Strengths:**

1) The paper reformulates 3D point cloud registration as a diffusion and denoising framework, and the results validate the effectiveness of the framework.

2) The paper is fairly well-written and easy to follow.

**Weaknesses:**

1) The title of the paper says the method is "robust". However, it seems there is a dilemma in sample diversity and sample efficiency for the proposed method. That means a fixed set of hyper-parameters might not work well for different kinds of datasets. In other words, the method is not as robust as claimed in the title.

2) Only mAP is used for evaluating 6D pose estimation, other popular evaluation metrics such as ADD and the BOP metrics should be considered for a more comprehensive understanding of the method's performance.

3) [a] also learns a SE(3) diffusion model. Maybe it should be discussed as a related work.

[a] SE(3)-DiffusionFields: Learning smooth cost functions for joint grasp and motion optimization through diffusion. ICRA 2023.

**Questions:**

1) Could the authors provide the training time cost comparison of the proposed method and the compared methods? Does it require significantly more time for training compared to existing methods?

2) How about performing the interpolation and perturbation of the transformation on so(3) + R^3 (which is also a linear vector space) instead of se(3)?

**Limitations:**

Yes.

---

> ### Author Rebuttal · Authors · 2023-08-09
>
> **Q1**: The title of the paper says the method is "robust". However, it seems there is a dilemma in sample diversity and sample efficiency for the proposed method. That means a fixed set of hyper-parameters might not work well for different kinds of datasets. In other words, the method is not as robust as claimed in the title.
>
> **A1**: Thank you for your insightful comments and constructive feedback. The “robust” claimed in the title mainly refers to that our SE(3) diffusion registration models can present excellent estimation robustness on real-world pose estimation challenges (e.g., full-range transformation and severe occlusions). Additionally, Table 2 shows that a fixed set of hyper-parameter settings can consistently bring significant performance gain across different datasets, which strongly supports the algorithm robustness of our method.
>
> **Q2**: Only mAP is used for evaluating 6D pose estimation, other popular evaluation metrics such as ADD and the BOP metrics should be considered for a more comprehensive understanding.
>
> **A2**: Thanks for your suggestion. We will add more evaluation metrics (such as ADD and the BOP metrics) in our revised version. In this rebuttal response, we give the results of methods DCP, Diff-DCP, RPMNet and Diff-RPMNet over ADD, and four BOP metrics (MSSD, AD, ADI, PROJ) on LINEMOD benchmark as below. It can be observed that our SE(3) diffusion registration methods (Diff-DCP and Diff-RPMNet) can consistently outperform their baseline methods (DCP and RPMNet) by a large margin on all metrics. We will add these comparison results in our revised version.
>
> |     |  ADD | MSSD  | AD  | ADI  | PROJ |
> |-----|---|---|---|---|---|
> | DCP |  0.226 |	0.378|	0.430|	0.893|	0.021
> | Diff-DCP |  0.640|	0.739|	0.801|	0.967|	0.340
> | RPMNet | 0.341| 0.417|	0.521|	0.897|	0.071
> | Diff-RPMNet|  0.609|	0.710|	0.779|	0.959|	0.267
>
>
>
> **Q3**: [a] also learns a SE(3) diffusion model. Maybe it should be discussed as a related work.
>
> **A3**: Thanks for pointing it out. [a] proposes an SE(3) diffusion model for robotic grasping tasks and presents promising results. Since it’s a concurrent paper with us, we didn’t notice and discuss it in our original paper. We will discuss it as a related work in our revised version.
>
> **Q4**: Training time cost comparison of the proposed method and the compared methods? Does it require significantly more time for training?
>
> **A4**: We hope to clarify that to ensure a fair comparison, we take the same training epochs as compared methods for performance evaluations. Particularly, during each epoch, given a training sample {X, M, H_0}, the compared methods directly use it for training, while our method generates one diffusion sample {X_t, M, H_t} for training (please refer to Alg. 1). Thus, their numbers of training samples utilized in each epoch are also identical. This indicates that our diffusion model shares a comparable training time with other methods and it doesn't require significantly more time for training.
>
> **Q5**: How about performing the interpolation and perturbation of the transformation on so(3) + R^3 instead of se(3)?
>
> **A5**: As shown in table below, compared to se(3) diffusion, so(3)+R^3 diffusion presents some performance degradation on TUD-L dataset. Their performance differences primarily stem from the disparity in quality of training samples generated by se(3) diffusion and so(3)+R^3 diffusion. Specifically, se(3) diffusion is demonstrated to generate smooth and shortest interpolation path (i.e., geodesic path) between two transformations, which can provide high-quality training samples to train the denoising network for a more reliable se(3) reverse process (Please refer to Alg. 1). Instead, so(3)+R^3 diffusion trace would generate non-smooth and unstable diffusion path, which would generate large amounts of low-quality training samples and thereby degrade the pose-correction ability of the trained denoising network.
>
> |   | 5$^{\circ}$@mAP|10$^{\circ}$@mAP |1cm@mAP|2cm@mAP|
> |-----|---|---|---|---|
> | Diff-RPMNet w/ so(3)+R^3 | 0.63 |0.93|0.95|0.98
> | Diff-RPMNet w/ se(3) | **0.90**|**0.98**|**0.98**|**0.99**

---

> > ### Comment · Reviewer_9j1P · 2023-08-19
> >
> > Thanks for the clarifications.

---

### Official Review · Reviewer_NS5b · 2023-06-30

**Soundness:** 3 good
**Presentation:** 3 good
**Contribution:** 3 good
**Rating:** 8
**Confidence:** 3

**Summary:**

The authors propose an approach to address the problem of 6D pose estimation on real-world data based on a denoising diffusion process.
They introduce a novel diffusion process on the SE(3) manifold, leveraging Lie algebra se(3) to shift the process from the linear Euclidean to the nonlinear SE(3) space.
Furthermore, a registration-specific variational lower bound is derived as optimization objective for model-based learning.
By reformulating the transformation following the posterior of the denoising matching term, the prior mean can be adapted to incorporate a surrogate registration model, enabling the application of existing deep registration models.
The experimental results show that not all registration models can deal with real-world data and emphasize the significant effectiveness of the novel SE(3) diffusion process.


**Strengths:**

Originality:
A novel diffusion process on the nonlinear SE(3) space for point cloud registration is introduced.

Quality:
The paper is well written in terms of language and organization, and a solid mathematical foundation for the proposed diffusion process is provided.
The extensive experiments highlight the superior performance of the approach compared to the respective baselines and a variety of other related work.
The ablation studies demonstrate the impact of the approach and the influence of different configurations on multiple datasets.

Clarity:
Despite the emphasis on the mathematical background, all mathematical derivations are explained concisely and comprehensibly.
The figure of the framework overview and the provided pseudo codes of the training and inference algorithms allowing to grasp the underlying concepts easily.

Significance:
The novel denoising diffusion process significantly outperforms the baselines on real-world data, showing the effectiveness of the approach compared to related work.


**Weaknesses:**

I could not find any major weaknesses. The authors:
- Explain the method well
- Conduct extensive experiments and compare their approach to multiple related works on different datasets (only real-world datasets, but this is intended)
- Provide quantitative and qualitative results and analyzed which part of their approach contribute to the improvements
- Include ablation studies of the diffusion process (forward & backward) to further demonstrate the impact of their approach
- The Appendix contains the mathematical derivations in detail, additional visualization, and a section on broader impact, limitations and future work


**Questions:**

Where do the evaluation results for the related methods in Table 1 come from?
Did you evaluate the methods yourself?
If so, which code base did you use, or where did you find these results?
Most methods do not provide them in their papers.

Why are only the qualitative results for RPMNet listed in the Appendix and not also for DCP?
Since DCP was used in the ablation studies, it would also be interesting to see some visual results of it.


Appendix A.1 $\rightarrow$ equation 5 $\rightarrow$ penultimate term: distribution $p$ is used in the numerator, but should it not be distribution $q$?

**Limitations:**

The paper addresses the broader implications in terms of societal and academic impact.
Moreover, it discusses the limitations of their work and indicated possible future directions based on these limitations.

---

> ### Author Rebuttal · Authors · 2023-08-09
>
> **Q1**: Where do the evaluation results for the related methods in Table 1 come from? Did you evaluate the methods yourself? If so, which code base did you use, or where did you find these results? Most methods do not provide them in their papers.
>
> **A1**: Thanks for your positive score and encouraging comments. Due to the unavailability of source code for MN-DCP and MN-IDAM, we reference their reported experimental outcomes from the original papers for performance evaluation. The remaining results such as  FMR, RGM, RIENet, DCP, and RPMNet are evaluated by ourselves. To ensure a fair comparison, we employ their official codes for implementation and meticulously fine-tune them to enhance their performance on real-world 6D object pose estimation datasets.
>
> **Q2**: Why are only the qualitative results for RPMNet listed in the Appendix and not also for DCP? Since DCP was used in the ablation studies, it would also be interesting to see some visual results of it.
>
> **A2**: Thanks for your suggestion. We have presented some qualitative comparisons of DCP and Diff-DCP on TUD-L, LINEMOD and Occluded-LINEMOD datasets in Fig.1 of the uploaded rebuttal material (Please refer to the attached PDF file). In the future, following your suggestion, we will add these visualization results into our qualitative evaluations in our revised version.
>
> **Q3**: In equation 5 of Appendix A.1, distribution p is used in the numerator, but should it not be distribution q?
>
> **A3**: Thanks for pointing this typo out. We will correct it by replacing "p" with "q" in our revised paper.

---

> > ### Comment · Reviewer_NS5b · 2023-08-16
> >
> > Thanks for those clarifications.

---

### Official Review · Reviewer_WNiN · 2023-07-03

**Soundness:** 4 excellent
**Presentation:** 3 good
**Contribution:** 3 good
**Rating:** 7
**Confidence:** 4

**Summary:**

This paper introduces a point cloud alignment method based on a diffusion model on SE(3). For this, the forward and reverse diffusion processes are performed in the lie group se(3). The method is evaluated on challenging real datasets, showing significant improvements over its baselines.

**Strengths:**

Relevant Application and Challenging Method: The paper addresses the highly relevant topic of 6DoF pose estimation using diffusion models, which is an area of significant interest. Applying diffusion models to complex parameter spaces like SE(3) poses a challenging problem that has yet to be fully resolved.

Experimental setup: The method is evaluated on challenging real datasets that are often used in the literature regarding 6DoF pose estimation and refinement. The comparison with a significant number of existing methods is fair and comprehensive. Multiple scenarios and thresholds are evaluated, allowing for a better insight into the strengths and weaknesses of the different methods.

Results: The results validate the method, showing significant improvements over the baseline methods. Several ablation studies allow for insights into the method's parameterizations.

Method: The method can be used with different backbones, increasing its impact as an add-on method that potentially strengthens any existing baseline.

Writing: The paper is well written and guides the reader through the traditionally notation-heavy math of the diffusion model. The ideas are well motivated and explained.

**Rebuttal** The reviewer appreciates the clarifications and additional experiments provided in the rebuttal and maintains the initial rating.

**Weaknesses:**

Some related work on diffusion models over SE(3) / SO(3) could be cited. Examples include:

[R2] Leach et al., Denoising diffusion probabilistic models on so(3) for rotational alignment, ICLRW 2022.
[R3] Urain et al., SE(3)-DiffusionFields: Learning smooth cost functions for joint grasp and motion optimization through diffusion, ICRA 2022.

**Questions:**

Q1: 179: There are several different ways for interpolating rotations; I've often seen quaternions used for this. Is there a motivation for using the exponential map over other representations? Similarly for the perturbations.

Q2: 268: How are 512 points sampled if the objects are smaller than that (especially in real scenes)?

(Suggestion for future work, no effect on rating, no need to respond): The authors might want to look at SYMSOL [R1], data synthetic dataset especially for dealing with pose ambiguities.

[R1] Murphy, Kieran A., et al. "Implicit-PDF: Non-Parametric Representation of Probability Distributions on the Rotation Manifold.", ICML 2021

**Limitations:**

Limitations and impact are adequately discussed.

---

> ### Author Rebuttal · Authors · 2023-08-09
>
> **Q1**: 179: There are several different ways for interpolating rotations; I've often seen quaternions used for this. Is there a motivation for using the exponential map over other representations? Similarly for the perturbations.
>
> **A1**: Thanks for your valuable and positive comments. (1) There are indeed some methods for interpolating 3DoF rotations, such as using quaternion or matrix representations. However, our task needs to employ more complex 6DoF transformation interpolation. In this context, the utilization of the 6DoF exponential map for transformation interpolation has been demonstrated to generate smooth and shortest interpolation path (i.e., geodesic path) between two transformations. Instead, decoupling 6DoF transformation interpolation into separate 3DoF rotation interpolation and 3DoF translation interpolation would suffer from non-smooth, unstable interpolation traces, resulting in the low-quality training sample for denoising-network training (see Alg. 1). The experimental results on TUD-L dataset in table below also empirically validates that using 6DoF exponential map for transformation interpolation can achieve higher estimation precisions than the 3DoF quaternion+3DoF translation interpolation. Therefore, we prefer using the exponential map over other representations for transformation interpolation.
>
> (2) For 6DoF transformation perturbations, both 6DoF exponential-map perturbation and 3DoF quaternion+3DoF translation perturbation are valid options. In our paper, to maintain content coherence, we still employ the exponential map representation for transformation perturbation.
>
> |   | 5$^{\circ}$@mAP|10$^{\circ}$@mAP |1cm@mAP|2cm@mAP|
> |-----|---|---|---|---|
> | Diff-RPMNet w/ 3DoF quaternion+3DoF translation | 0.63 |0.93|0.95|0.98
> | Diff-RPMNet w/ 6DoF exponential map | **0.90**|**0.98**|**0.98**|**0.99**
>
>
> **Q2**: 268: How are 512 points sampled if the objects are smaller than that (especially in real scenes)?
>
> **A2**: If the number of scanned object points is less than 512, some points will be sampled repeatedly to meet the required point count. For example, if we want to sample a point set of size 5 from 3 points {p1, p2, p3}, the resulting sampled points would be {p1, p1, p2, p2, p3}, where points {p1, p2} are sampled repeatedly.
>
> **Q3**: Some related work on diffusion models over SE(3) / SO(3) could be cited.
>
> **A3**: Thanks for pointing them out. We will incorporate them into our related work section for a thorough discussion in the revised version.

---

> > ### Comment · Reviewer_WNiN · 2023-08-17
> >
> > Thank you for the additional details and clarifications.

---

### Official Review · Reviewer_VaQE · 2023-07-05

**Soundness:** 1 poor
**Presentation:** 2 fair
**Contribution:** 2 fair
**Rating:** 4
**Confidence:** 5

**Summary:**

This paper proposes a SE(3) diffusion model-based point cloud registration framework for robust 6D object pose estimation, which formulates point cloud registration as a denoising diffusion process and enables progressive refinement of the transformation between the source point cloud and the model point cloud. Experiments are conducted on TUD-L, LINEMOD, and Occluded-LINEMOD datasets, and state-of-the-art results are achieved on these datasets.

**Strengths:**

1. The paper is overall well written and easy to follow.
2. Experiments are conducted on three different datasets and comparisons are done to different methods. High performance is achieved.


**Weaknesses:**

1. Since this paper aims for 6D object pose estimation, the compared methods should mainly focus on 6D object pose estimation methods obviously, not just some point cloud registration methods.
2. The proposed method is trained and tested on 6D object pose estimation datasets, However, the compared methods are point cloud registration methods. Did the authors retrain these learning-based point cloud registration models on the datasets in order to fairly compare with these learning-based methods? I am concerned that this is the key to the improved performance.
3. The comparisons are done to a few methods in the current manuscript, I suggest more state-of-the-art methods should be included for comparison, e.g., point cloud registration methods (Predator[1], Cofinet[2], Geotransformer[3], and so on) and 6D object pose estimation (Fs-net[4], Ove6d[5], Gpv-pose[6] and so on).
[1] Shengyu Huang, Zan Gojcic, Mikhail Usvyatsov, Andreas Wieser, and Konrad Schindler. Predator: Registration of 3d point clouds with low overlap. In CVPR, 2021.
[2] Hao Yu, Fu Li, Mahdi Saleh, Benjamin Busam, and Slobodan Ilic. Cofinet: Reliable coarse-to-fine correspondences for robust point cloud registration. NeurIPS, 2021.
[3] Zheng Qin, Hao Yu, Changjian Wang, Yulan Guo, Yuxing Peng, and Kai Xu. Geometric transformer for fast and robust point cloud registration. In CVPR, 2022.
[4] Wei Chen, Xi Jia, Hyung Jin Chang, Jinming Duan, Shen Linlin, and Ales Leonardis. Fs-net: Fast shape-based network for category-level 6d object pose estimation with decoupled rotation mechanism. In CVPR, 2021.
[5] Dingding Cai, Janne Heikkila, and Esa Rahtu. Ove6d: Object viewpoint encoding for depth-based 6d object pose estimation. In CVPR, 2022.
[6] Yan Di, Ruida Zhang, Zhiqiang Lou, Fabian Manhardt, Xiangyang Ji, Nassir Navab, and Federico Tombari. Gpv-pose: Category-level object pose estimation via geometry-guided point-wise voting. In CVPR, 2022.


**Questions:**

See Weaknesses.

**Limitations:**

The authors mention the limitation of the proposed method in supplementary material.

---

> ### Author Rebuttal · Authors · 2023-08-09
>
> **Q1**: Compared methods should focus on 6D object pose estimation methods obviously, not just point cloud registration methods.
>
> **A1**: Thanks for your diligent comments to help improve our work. Considering that the RGB(D)-based pose estimation methods would suffer from limited robustness to challenging lighting conditions (e.g., low-light conditions or color-varying light conditions), our work thus focuses on how to utilize geometry-rich point-cloud data (without any RGB information) to achieve lighting-robust, instance-level pose estimation. Based on this task setting, point cloud registration methods are the suitable and popular 6D pose estimation methods, where the orientation and position of the object can be effectively recovered by registering the scanned object point cloud to the model one.
> However, the conventional RGB(D)-based methods are unable to adapt to this setting due to the requirement for additional RGB information. Therefore, to ensure a fair comparison, following [R1], we compare our method with those registration-based pose estimation methods rather than conventional RGB(D)-based methods. In the future, following your suggestion, we will add some RGB(D)-based 6D pose estimation methods (e.g., your listed papers below) for performance comparisons in our revised version.
>
> [R1]: Learning-based Point Cloud Registration for 6D Object Pose Estimation in the Real World. ECCV’2022
>
> **Q2**: Did the authors retrain these learning-based point cloud registration models on the datasets in order to fairly compare with these learning-based methods? I am concerned that this is the key to the improved performance.
>
> **A2**: To ensure a fair comparison, all compared learning-based point cloud registration models are retrained on the training data of 6D object pose estimation datasets for performance comparisons. Particularly, we employ their official codes for implementation and meticulously fine-tune them to enhance their performance on real-world 6D object pose estimation datasets. It should be noted that due to the unavailability of source code for MN-DCP and MN-IDAM, we reference their reported experimental outcomes from the original papers for performance comparisons.
>
> **Q3**: The comparisons are done to a few methods in the current manuscript, I suggest more state-of-the-art methods should be included for comparison.
>
> **A3**: Thanks for your suggestion. We will incorporate the methods you've mentioned for comparisons in our future version. Furthermore, we hope to clarify that the listed registration methods [1, 2, 3] primarily center around scene-level registration rather than object-level. Due to the huge scale difference between the scene and object point clouds, adapting their network architecture and hyperparameters to effectively extend them for object-level registration and object pose estimation is a non-trivial task. Hence, our initial paper version does not engage in a performance comparison with them. In the future, we will make our best effort to fine-tune their network architecture and hyperparameters to better suit object-level registration for pose estimation. Additionally, we note that among these methods, Predator [1] offers a set of hyperparameters specifically for object-level registration, and we retrain it on the TUD-L dataset for comparisons. The table below shows that our method can achieve significant performance advantage over Predator with a lower time cost.
>
>   Moreover, considering that the RGB(D)-based instance-level pose estimation methods still suffer from limited robustness to challenging lighting conditions (e.g., low-light conditions or color-varying light conditions), our paper thus focuses on improving the lighting robustness of instance-level methods via point cloud registration. Hence, we mainly compare our method with the instance-level methods rather than the categorical methods as you listed. In our future version, we will include your suggested methods into our experimental comparisons.
>
> |     |  5$^{\circ}$@mAP | 10$^{\circ}$@mAP  | 1cm@mAP  | 2cm@mAP  | Times (Sec.)
> |-----|---|---|---|---|---|
> | Predator |  0.42|	0.71|	0.66|	0.81|	0.62 | 0.62
> | Diff-DCP (ours) |  0.65 |	0.85 |	0.73|	0.94|	**0.13**
> | Diff-RPMNet (ours) |  **0.90**|	**0.98**|	**0.98**|	**0.99**| 	0.17

---

> > ### Comment · Reviewer_VaQE · 2023-08-19
> >
> > Thank you for the author's feedback.
> > This paper is dedicated to the task of point cloud-based 6D object pose estimation, and there are so many depth-only-based 6D object pose estimation methods, e,g, OVE6D, StablePose, CloudAAE, CloudPose, and so on. I think it is necessary to include these methods in the comparisons. Otherwise, I worry that the proposed method is not convincing enough.
> > I think a thorough comparison with existing methods is necessary to verify the effectiveness of the proposed methods. Furthermore, the listed point cloud registration methods also could perform well on object-centric datasets. Besides, more state-of-the-art object-level point cloud registration methods have emerged recently, and comparison with the latest state-of-the-art methods is necessary.

---

> > > ### Author Response · Authors · 2023-08-21
> > >
> > > We thank the reviewer for reviewing and replying to our rebuttal.
> > > *  Regression-based vs Matching-based Methods: We note that most previous depth-only-based methods (e.g., CloudAAE, CloudPose, and StablePose) on 6D pose estimation are based on direct pose **regression**, disregarding the utilization of object models for pose reference. As a result, these methods face limitations in their ability to generalize to unseen objects, primarily due to the absence of model-informed pose references.  Although OVE6D exhibits the potential to generalize to unseen objects, it necessitates the additional encoding of new instances into the viewpoint codebook, which could potentially increase its application costs. By contrast, our method follows a drastically different research trajectory that leverages **3D matching** between the scanned object point cloud (PC) and the model PC for pose estimation. Therefore, its inherent model generalization ability in 3D matching potentially facilitates the pose estimation of unseen objects. To validate it, we choose objects with IDs=1~8 from LINEMOD to train our Diff-DCP and test its generalization performance on the unseen object with ID=9. Also, we directly generalize the Diff-DCP trained on LINEMOD to the TUD-L object with ID=3. We report the seen results vs unseen results in the table below. We observe that our method Diff-DCP can still achieve meaningful and encouraging estimation precisions on unseen objects without any additional model fine-tuning.
> > >
> > >    |     |LINEMOD (ID=9) |	TUD-L (ID=3)|
> > >    |:-----|:-----|:-----|
> > >    | | mAP (5&deg;/10&deg;/1cm/2cm)|mAP (5&deg;/10&deg;/1cm/2cm)|
> > >    Diff-DCP (seen) |	0.12/0.46/0.80/0.96	| 0.92/0.98/0.89/0.97|
> > >    Diff-DCP (unseen) |	0.10/0.39/0.68/0.82 |	0.53/0.67/0.57/0.65|
> > >
> > > * Core Contribution to Matching-based Methods: As mentioned in lines 30-41, current object-level registration methods (e.g., MN-DCP in ECCV’2022) still suffer from limited matching robustness for pose estimation due to real-world challenges (e.g., full-range rotation and severe occlusion). It promotes us to develop an SE(3) diffusion registration model as a **plug-and-play** method to advance current registration methods for more reliable pose estimation. As a general framework, our method can integrate with different deep registration models to boost their performance (Ref: DCP vs Diff-DCP and RPMNet vs Diff-RPMNet in Table 1), thereby increasing our impact as an add-on method to enhance existing registration models. Theoretically, we can also include Predator, GeoTransformer and CoFiNet as our surrogate registration models (see Eq.12) into our framework for their performance boosting. We are highly interested to validate it in our future paper version. Also, we think the significant performance improvements achieved by Diff-DCP and Diff-RPMNet over their baselines (see Table 1) have sufficiently verified the effectiveness of our framework.
> > >
> > > * Summary: Different from most previous **regression**-based depth-only-based methods, our method is based on **3D matching**, enabling our method to enjoy the promising generalization ability to unseen object pose estimation. Furthermore, our SE(3) diffusion registration model is a general, **plug-and-play** framework. We can take different deep registration methods (such as the listed Predator and GeoTransformer) for their performance improvement. Finally, we regret that due to time constraints, we have to perform **a thorough comparison** with existing methods, e.g. depth-only-based methods and more point cloud registration methods, in our final paper version.
> > >
> > > Finally, we sincerely hope the reviewer will reconsider our paper for following aspects: (i) We pioneer the SE(3) diffusion registration model for robust 6D object pose estimation, with two innovative components: a transformation interpolation-based SE(3) diffusion process and a surrogate registration-driven SE(3) reverse process; (ii) In this context, we derive an effective registration-specific variational lower bound for model optimization (Detailed proof can be found in Appendix A.1); (iii) Our method is a general, plug-and-play framework, which can effectively integrate with different deep registration models to boost their performance, significantly increasing our impact as an add-on method to enhance existing registration models; (iv) Extensive experiments on real-world 6D object pose estimation datasets validate that our diffusion variants (Diff-DCP and Diff-RPMNet) consistently achieve significant performance improvements over their baseline models (DCP and RPMNet).

---

### Official Review · Reviewer_5ZGT · 2023-07-06

**Soundness:** 3 good
**Presentation:** 3 good
**Contribution:** 3 good
**Rating:** 5
**Confidence:** 3

**Summary:**

The paper proposes and SE(3) diffusion model for pose estimation i.e. rotation R and translation t for the use case of registration two pointclouds. The paper addresses an important problem in 3D vision and which has applications in Robotics, AR/VR. The author claim that the convential diffusion models won't work for this task of SE(3) diffusion and proposes several changes to constaint the transformation transitions during the diffusion and reverse processes. Results are shown on competing and challenging baselines.

**Strengths:**

The technical contributions look sound and the author addresses an important and challenging problem in 3D vision. Adapting conventional diffusion models to SE(3) using a thorough mathematical formulation is convincing. Results show strong improvement on quantitative metrics.

**Weaknesses:**

1. I wonder why the authors didn't compare or show experiments on more challenging categorical 6D pose and size estimation benchmarks nor discuss those in related works [1][2][3]. Is it a limitation of current work? This should at least be discussed in literature review for the work to have more merit since conventional instance-based pose estimation which assumes known CAD models has mostly been solved achieving higher accuracies. Might this be one of the reasons the author gets smaller improvements in some of the metrics?

2. Relating to my point above, the author only evaluates their approach on a single and very simpler benchmark. I understand this is what the authors are proposing i.e. point cloud registration but adding more benchmarks (within the point cloud registration works such as in the wild pose estimation etcetera) could further reinforce the author's results since it is hard to fully validate the efficacy of the results when evaluated on a single benchmark. Recently, the shift has been more toward the wild pose estimation[4] and zero-shot pose estimation and I wonder if this work can directly be useful to some of the other more complicated problems as well.

3. One of the pain points of traditional ICP is parameter tuning which can improve results although takes a lot of hand-tuning. Does the author's approach suffer from the same problem? like a number of denoising steps and etc.

4. Diffusion models are slow to train and infer. Can the authors comment on their inference speeds vs accuracy i.e. if they use a smaller denoising timestep? In essence, I saw no comparisons to state-of-the-art timing results for 6D pose estimation i.e. [2][3], or a direct discussion of it. Timing is crucial for robotics applications and the work's strong improvement might be downplayed if this is very slow and not real-time.


[1] Normalized Object Coordinate Space for Category-Level 6D Object Pose and Size Estimation, Wang et al.
[2] CenterSnap: Single-Shot Multi-Object 3D Shape Reconstruction and Categorical 6D Pose and Size Estimation Irshad et al.
[3] ShAPO : Implicit Representations for Multi Object Shape Appearance and Pose Optimization, Irshad et al.
[4] Category-Level 6D Object Pose Estimation in the Wild: A Semi-Supervised Learning Approach and A New Dataset, Fu et al.

**Questions:**

Please see the remarks in the weakness section. I am willing to improve my score if the authors address my concerns about 1. more complicated benchmarks i.e comparisons to categorical 6D pose and size estimation or in the wild pose estimation, discussion/comments about parameter tuning for the author's approach, discussion/comparison to timing i.e. speed vs accuracy tradeoff relation to some of the state of the art, fast 6D pose and size estimation approaches.

**Limitations:**

The author's work focuses on point cloud registration only and requires a depth map or depth sensor and wouldn't work in a monocular i.e. RGB only case

---

> ### Author Rebuttal · Authors · 2023-08-09
>
> **Q1**: Instance-based pose estimation has been solved? Why not compare categorical benchmarks?
>
> **A1**: Thanks for your valuable comments to help improve the quality of our paper.
>
> (1) Although the current RGB(D)-based instance-level pose estimation methods have achieved relatively good performance, they usually require the assumption of access to high-quality RGB images. As such, in more challenging lighting conditions (e.g., low-light conditions or color-varying light conditions), their instance-level estimation precisions would be significantly degraded. As shown in the table below, when we use the Gamma transformation (gamma=5,7,9) to darken the RGB image, the estimation precisions of the SOTA GDR-Net would degrade significantly. Thus, realizing lighting-robust, instance-level pose estimation is still an unresolved and meaningful task for wider applications of 6D pose estimation.
>
> (2) To alleviate the aforementioned dilemma, our paper thus focuses on how to utilize geometry-rich point-cloud data (without any RGB information) to achieve lighting-robust, instance-level pose estimation. The table below verifies that our method presents excellent robustness to challenging lighting conditions.
> However, as our method exploits the object-model alignment for instance-level estimation, the inherent intra-class shape variation in category-level estimation would confuse such alignment and thereby reduce the pose accuracy. As such, our current method mainly focuses on instance-level datasets for evaluation, rather than categorical datasets. In the future, following your suggestion, we will add discussions of papers [1, 2, 3] in our related work.
>
> Moreover, by integrating NOCS [R1] (normalized object coordinate space) into our model, the NOCS-enhanced SE(3) diffusion registration framework has the potential to address category-level estimation tasks. Specifically, in the surrogate registration model, we can take NOCS as the category-level model point cloud (PC), which can be used to construct category-level correspondences with the source PC for pose and scale estimations. This enables our SE(3) reverse process to gradually denoise the object pose and scale in a category-level setting. It's noted that our SE(3) diffusion process can directly adapt to category-level tasks without any modification.
>
> [R1] Normalized Object Coordinate Space for Category-Level 6D Object Pose and Size Estimation, CVPR’2019.
>
> |   | 5$^{\circ}$@mAP|10$^{\circ}$@mAP |1cm@mAP|2cm@mAP|
> |-----|---|---|---|---|
> | GDR-Net (Gamma=5) | 0.10 |0.23|0.06|0.14
> | GDR-Net (Gamma=7) | 0.02 |0.08|0.02|0.05
> | GDR-Net (Gamma=9) | 0.01 |0.04|0.01|0.02
> | Diff-RPMNet (ours) | 0.18|0.47|0.51|0.72
> | Diff-DCP (ours) |  **0.22**|**0.51**|**0.65**|**0.82**
>
> **Q2**:  Only evaluate on a single and very simpler benchmark? Is our method useful for wild or zero-shot pose estimation?
>
> **A2**: (1) We hope to clarify that our method was not solely evaluated on a single and simpler benchmark. Instead, following the SOTA method [R2], we conducted extensive comparisons on three challenging and widely-used 6D estimation benchmarks (TUD-L, LINEMOD, and Occluded-LINEMOD). Although they are instance-level datasets, using pure point-cloud data to achieve lighting-robust, instance-level estimation is still an unresolved yet important problem for wider applications of 6D pose estimation.
>
> (2) It’s a highly interesting idea to extend our framework to wild/zero-shot pose estimation.
> Since our current model focuses on using 3D registration to realize lighting-robust, instance-level estimation, we need to make some modifications for enabling our model to handle wild/zero-shot estimations. Specifically, inspired by NOCS [R1], we can further exploit the vector quantization to compress the different instance models into a set of primitive keypoint representations. As such, these primitive representations can be viewed as the primitive model for constructing correspondences with the unseen object PC to estimate its pose and scale.
>
> [R2]: Learning-based Point Cloud Registration for 6D Object Pose Estimation in the Real World. ECCV’2022
>
> **Q3**: Whether our method suffer from a lot of hand-tuning, such as the number of denoising steps, like in ICP?
>
> **A3**: Our framework can achieve good performance without a lot of hand-tuning like ICP. In contrast, our ablation studies in Table 2 show that different configurations can consistently bring significant performance gains over the baseline. Particularly, for the number of denoising steps, Fig.3 shows that more steps bring lower errors across all datasets. These ablation results effectively validate the robustness of our method to these hyperparameters, and indicate that our method can be free from heavy hand-tuning like ICP for achieving good performance.
>
> **Q4**: Diffusion models are slow to train and infer? Comment on their inference speeds vs accuracy?
>
> **A4**: (1) For training time, to ensure a fair comparison, we take the same training epochs as compared methods for model training. Particularly, during each epoch, given a training sample {X, M, H_0}, the compared methods directly use it for training, while our method generates one diffusion sample {X_t, M, H_t} for training (please refer to Alg. 1). Thus, their numbers of training samples utilized in each epoch is identical. This indicates that our diffusion model shares a comparable training time with other methods and thus our diffusion model is not slow to train.
>
> (2) For inference time, Fig.3 shows that it depends on the number of denoising steps, and a smaller timestep would accelerate the inference speed while decreasing the accuracy. For a good balance, we set denoising steps to 5, which can achieve excellent precisions with tolerable speed (~0.17s). In the future, as discussed in our future work, we will convert our SE(3) diffusion registration from the point-cloud space to the compact feature space for increasing our inference speed.

---

### Author Rebuttal · Authors · 2023-08-09

To address Q2 raised by Reviewer-NS5b, we have included some qualitative comparisons of DCP and Diff-DCP on TUD-L, LINEMOD, and Occluded-LINEMOD datasets in the attached PDF file.

---

### Decision · Program_Chairs · 2023-09-21

**Decision:**

Accept (poster)

**Comment:**

This paper obtained discordant ratings after the review stage. Certain reviewers appreciated the novelty of the point cloud registration approach proposed and the solidity of the results that it obtained. Some other reviewers highlighted the need for clarifications of certain methodological issues and the need for more experimental evaluations. Despite the discordant ratings were not completely resolved during discussion, a clear consensus emerged towards acceptance among the reviewers, with four out of five reviewers eventually suggesting leaning towards acceptance and only one reviewer sticking on a 4 rating. In particular, the comments of the reviewers who expressed a stronger signal for acceptance were deemed particularly meaningful by the AC. The final recommendation is thus to accept this paper.